# Nanocarriers Containing Curcumin and Derivatives for Arthritis Treatment: Mapping the Evidence in a Scoping Review

**DOI:** 10.3390/pharmaceutics17081022

**Published:** 2025-08-06

**Authors:** Beatriz Yurie Sugisawa Sato, Susan Iida Chong, Nathalia Marçallo Peixoto Souza, Raul Edison Luna Lazo, Roberto Pontarolo, Fabiane Gomes de Moraes Rego, Luana Mota Ferreira, Marcel Henrique Marcondes Sari

**Affiliations:** 1Curso de Biomedicina, Departamento de Análises Clínicas, Universidade Federal do Paraná, Curitiba 80210-170, Brazil; beatriz.sato@ufpr.br (B.Y.S.S.); susan.chong@ufpr.br (S.I.C.); 2Programa de Pós-Graduação em Ciências Farmacêuticas, Universidade Federal do Paraná, Curitiba 80210-170, Brazil; nathalia.marcallo@ufpr.br (N.M.P.S.); raulluna@ufpr.br (R.E.L.L.); pontarolo@ufpr.br (R.P.);

**Keywords:** joint inflammation, nanostructures, curcuminoids, solubility

## Abstract

**Background/Objectives:** Curcumin (CUR) is well known for its therapeutic properties, particularly attributed to its antioxidant and anti-inflammatory effects in managing chronic diseases such as arthritis. While CUR application for biomedical purposes is well known, the phytochemical has several restrictions given its poor water solubility, physicochemical instability, and low bioavailability. These limitations have led to innovative formulations, with nanocarriers emerging as a promising alternative. For this reason, this study aimed to address the potential advantages of associating CUR with nanocarrier systems in managing arthritis through a scoping review. **Methods:** A systematic literature search of preclinical (in vivo) and clinical studies was performed in PubMed, Scopus, and Web of Science (December 2024). General inclusion criteria include using CUR or natural derivatives in nano-based formulations for arthritis treatment. These elements lead to the question: “What is the impact of the association of CUR or derivatives in nanocarriers in treating arthritis?”. **Results:** From an initial 536 articles, 34 were selected for further analysis (31 preclinical investigations and three randomized clinical trials). Most studies used pure CUR (25/34), associated with organic (30/34) nanocarrier systems. Remarkably, nanoparticles (16/34) and nanoemulsions (5/34) were emphasized. The formulations were primarily presented in liquid form (23/34) and were generally administered to animal models through intra-articular injection (11/31). Complete Freund’s Adjuvant (CFA) was the most frequently utilized among the various models to induce arthritis-like joint damage. The findings indicate that associating CUR or its derivatives with nanocarrier systems enhances its pharmacological efficacy through controlled release and enhanced solubility, bioavailability, and stability. Moreover, the encapsulation of CUR showed better results in most cases than in its free form. Nonetheless, most studies were restricted to the preclinical model, not providing direct evidence in humans. Additionally, inadequate information and clarity presented considerable challenges for preclinical evidence, which was confirmed by SYRCLE’s bias detection tools. **Conclusions:** Hence, this scoping review highlights the anti-arthritic effects of CUR nanocarriers as a promising alternative for improved treatment.

## 1. Introduction

“Arthritis” refers to a wide range of arthritic disorders, often linked to degenerative conditions and autoimmune responses [1,2]. This condition is more prevalent in women, with its incidence rate increasing with age [3,4]. It is characterized by chronic inflammation affecting one or more joints, leading to symptoms like pain, swelling, stiffness, and limited mobility [1,2]. Arthritis includes over 100 different kinds, with osteoarthritis being the most common. Other varieties include rheumatoid arthritis (RA), psoriatic arthritis, and related autoimmune diseases [5,6]. RA is a chronic inflammatory autoimmune disease marked by bilateral joint destruction and systemic effects [7], affecting about 1 to 2% of the adult population globally [5]. Osteoarthritis (OA) is a chronic degenerative joint disease [8] that impacts approximately 9.6% of men and 18% of women aged over 60 years [9]. According to WHO, around 18 million people were living with RA in 2019 [10], while 528 million were affected by OA [11]. Despite the availability of various treatments, a definitive cure for arthritis is still absent [12]. Commonly used medications include nonsteroidal anti-inflammatory drugs (NSAIDs), opioids, steroids, disease-modifying antirheumatic drugs, and immunosuppressants. However, long-term use of these therapies is often limited by adverse effects, tolerance, and suboptimal efficacy in halting disease progression. The existing limitations highlight the critical necessity for developing safer and more effective therapeutic alternatives [13]. Therefore, there is a pressing imperative to allocate resources towards innovative treatment options that can significantly improve the quality of life for patients afflicted with these chronic conditions.

In the context of chronic disease management, turmeric derivatives exhibit protective effects through their anti-inflammatory properties. Curcumin (CUR) is the primary natural polyphenol from *Curcuma longa* [14] and has been identified as a potential therapeutic agent for alleviating symptoms of RA by reducing pain and swelling [15]. Moreover, CUR demonstrates therapeutic effects akin to those of ibuprofen in the management of knee osteoarthritis (OA) [16]. This efficacy can be linked to its anti-inflammatory mechanisms, which involve the downregulation of various inflammatory mediators [17], all while maintaining a favorable safety profile by minimizing adverse effects [18].

CUR can inhibit Nuclear Factor Kappa B (NF-kB), a transcription factor crucial in inflammatory responses, since it suppresses the expression of pro-inflammatory cytokines, including TNF-α, IL-6, and IL-1β. Furthermore, the antioxidant effects are due to its capacity to enhance the antioxidant response by reducing lipid peroxidation and maintaining elevated levels of superoxide dismutase and catalase. Therefore, CUR emerges as a promising therapeutic alternative for treating arthritis.

However, despite its numerous properties, this compound exhibits limited potential when used as an oral medicine due to its low bioavailability. This restriction arises from low aqueous solubility, intestinal permeability, physicochemical instability, and a high rate of metabolism and clearance from the body [19]. To overcome these constraints, the integration of CUR into nanocarrier systems has garnered significant interest in recent research. Nanoencapsulation presents numerous benefits, such as improved solubility and stability of CUR, along with enhanced specificity in targeting delivery systems and the ability to regulate the release kinetics of the active compound. Various nanotechnological platforms have been explored to improve CUR’s therapeutic efficacy in arthritis models. These approaches aim to overcome conventional delivery barriers and maximize CUR’s anti-inflammatory action at the site of inflammation. Different CUR formulations are being explored to overcome these challenges. For example, in a study conducted by Wang and collaborators (2021) [20], the oral bioavailability of liposomal bisdemethoxycurcumin (BDMC) was approximately 10 times higher than that of free BDMC, as demonstrated by cumulative release rates both in vivo and in vitro. It highlighted an enhancement in the sustained effectiveness of BDMC in its encapsulated form compared to its free counterpart. Moreover, it was noted that, in contrast to traditional antioxidants known for their poor stability, limited joint retention, and suboptimal drug delivery to cartilage, nano-based formulations exhibit the capability to safeguard chondrocytes against oxidative stress. These formulations enhance the pharmacological efficacy of CUR delivery to the cartilage in rat models, effectively reducing hypernociceptive responses and inflammatory markers [20,21,22,23,24,25].

The growing interest in CUR association with nanocarriers emphasizes the need for a comprehensive synthesis of existing evidence regarding their efficacy in arthritis management. As research shifts toward novel formulations designed to enhance CUR’s therapeutic applications, it becomes essential to consolidate the scientific literature that highlights the benefits of utilizing nanocarriers as a cutting-edge therapeutic approach. This scoping review seeks to assess the utilization of CUR and its derivatives across various nanocarrier systems, drawing insights from both preclinical and clinical investigations. To the best of our knowledge, no scoping review has meticulously compiled the existing literature on CUR-loaded nanocarriers specifically targeted for arthritis therapy.

## 2. Materials and Methods

This scoping review followed the guidelines set by the Joanna Briggs Institute [26], with data reported according to PRISMA for Scoping Reviews [27]. The study protocol is available at OSF: https://doi.org/10.17605/OSF.IO/SE47R. Additionally, the PRISMA-ScR Fillable Checklist for Scoping Reviews was provided with Appendix A [28].

### 2.1. Research Strategy

The research was conducted across the PubMed, Scopus, and Web of Science databases, based on articles written in Roman characters and without time restrictions (December 2024). Each database employed a specific search strategy. Descriptors related to CUR/derivatives, arthritis, and nanostructured systems were combined using the Boolean operators “OR” and “AND” (Appendix A). Additionally, a manual search was conducted by reviewing the reference lists of included articles.

### 2.2. Eligibility Criteria

Eligibility criteria were established based on the PCC (Population, Concept, and Context) acronym, exploring the conceptual framework centered on the combination of nanoparticles as a strategy to enhance the bioavailability of CUR for the treatment of arthritis. The population consisted of nanocarrier systems, with the concept exploring the use of CUR in the context of arthritis treatment. These elements guided the question: “What is the impact of the association of CUR or derivatives in nanocarriers in treating arthritis?”.

As inclusion criteria, in vivo preclinical investigations and studies on human subjects that were full-text manuscripts written in Roman characters and assessed the use of CUR or derivatives in nanocarriers for treating arthritis were selected. Articles consisting solely of in vitro and in silico studies, those addressing acute pain or other chronic pain conditions unrelated to arthritis, review articles, meta-analyses, case reports, conference abstracts, editorials, book chapters, and articles associating CUR with multiple active compounds within the nanocarrier were excluded. Studies exploring intentionally chemically modified CUR were also not eligible.

### 2.3. Study Selection

The article selection process followed a two-stage protocol. First, an investigator conducted systematic searches across the specified databases. All studies retrieved from different databases were imported into Rayyan (Rayyan Systems, Inc., Doha, Qatar, free version), where duplicates were removed [29]. In the first stage, only two independent reviewers evaluated titles and abstracts blindly. Articles that met the inclusion criteria at this stage were subjected to a thorough full-text review to determine their eligibility for inclusion in the review. If there was any disagreement between reviewers, a third reviewer was consulted to resolve the issue. Articles not meeting the eligibility criteria were excluded from further consideration, with detailed documentation of the reasons for exclusion. The identification, selection, and inclusion process results were presented in a flowchart, which was made available on the PRISMA-ScR platform (Figure 1). This flowchart outlined the steps of identification, selection, eligibility criteria, and the outcomes (number of studies included and excluded) from the article studies.

### 2.4. Data Extraction and Synthesis

The next step involved investigators systematically compiling relevant data from the selected articles to create a comprehensive table using Microsoft Office Excel^®^ spreadsheets. This table included key information such as author names, publication year, nanocarrier type, formulation format, composition specifics, model types used in preclinical and clinical investigations, and a detailed account of pharmacological and toxicological assessments.

Qualitative data were systematically categorized according to predefined variables, including formulation type, CUR source, experimental model, and reported outcomes, to enable a thorough descriptive statistical analysis. An evidence map was constructed using Sankey plots, which are particularly effective for visualizing the distribution and interrelationships among these variables. The selection of Sankey diagrams was driven by their capacity to convey both the magnitude and directionality of connections across multiple categories. This visualization technique provides clear insights into the distribution of evidence across various combinations of variables, illustrating, for instance, the specific formulations evaluated within certain experimental models and their associated outcomes. Furthermore, using Sankey plots highlights clusters where evidence is concentrated, as well as gaps where certain combinations remain underexplored. Data analysis and the creation of the Sankey plots were performed using Microsoft^®^ Excel for the initial data organization and custom Python scripts (version 3.11.4), leveraging the Plotly library (version 5.18.0) for dynamic visualization (Appendix A).

### 2.5. Risk of Bias Evaluation

Two tools were applied to evaluate the risk of bias of the included studies: SYRCLE for animal studies [30] and RoB 2 [31]. Two reviewers independently conducted the quality assessment of the studies.

The SYRCLE Risk of Bias Tool is designed for evaluating animal studies (*n* = 31) and assesses six critical domains: selection, performance, detection, attrition, reporting, and other potential biases. It incorporates specific signaling questions to improve clarity, accuracy, and transparency in the assessment process. Each domain is classified as Yes, No, or Unclear, which corresponds to a low risk of bias, a high risk of bias, or insufficient information to ascertain the risk, respectively. In contrast, RoB 2 evaluates the risk of bias of randomized controlled trials (RCT (*n* = 3)) across five domains: randomization process, deviations from intended interventions, missing outcome data, measurement of the outcome, and selection of the reported result. Its categories for bias risk are “low”, “high”, or “some concerns”.

## 3. Results

Detailed information on the identification, screening, exclusion, and inclusion processes is provided in Figure 1. This scoping review identified 536 articles across three databases (PubMed: *n* = 75; Scopus: *n* = 304; and Web of Science: *n* = 157). After removing 190 duplicate records, 346 articles remained for the screening phase. Of these, 296 were excluded based on title and abstract screening, leaving 50 articles for full-text review. Fourteen studies were excluded due to inappropriate study design, one due to lack of access, and one due to an inappropriate study type, totaling 16 exclusions (Appendix A). Consequently, 34 articles were included in this review. A manual search was also conducted, identifying one additional article meeting the inclusion criteria; however, it was excluded due to lack of access to the full text.

**Figure 1 pharmaceutics-17-01022-f001:**
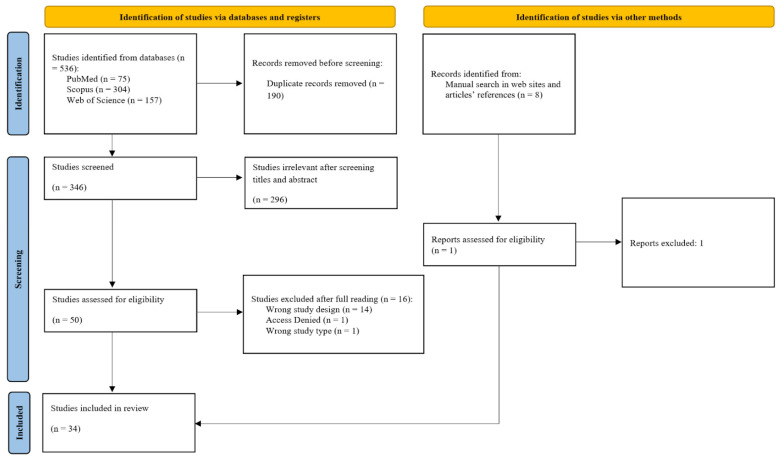
Scoping review flowchart. Source: PRISMA-ScR [27].

The tables present a chronological overview of the 34 studies’ key characteristics, focusing on their general physicochemical properties (Table 1) and experimental designs (Table 2). Each document investigates nano-based formulations aimed at modulating the effects of CUR for arthritis management, offering insights into the temporal trends in research concerning CUR-loaded nanocarriers for this purpose.

Most of the articles included in this review were published in 2024 (9/34) and 2023 (5/34); the remaining publications ranged from 2022 to 2012, indicating a recent trend in scientific literature. Geographically, China was the country with the highest number of included articles (12/34), followed by India (6/34); Pakistan (4/34); Iran (3/34); South Korea and the United States (2/34 each); and Brazil, Egypt, Spain, and Nigeria, each with 1 included study.

Regarding the descriptive evaluation, Figure 2A illustrates the distribution of organic versus inorganic nanocarriers across the reviewed studies, highlighting that the majority comprises organic materials (30/34). Regarding the types of nanocarriers containing CUR (Figure 2B), nanoparticles were the most employed system (17/34; ~50%), followed by nanoemulsions (4/34), micelles (3/34), and liposomes (2/34). Other types of nanocarriers were mentioned in only one study each. The association between CUR and nanocarriers resulted in three final forms of application (Figure 2C), with the liquid form being the most reported (23/34; ~68%; suspension), while the semisolid (7/34—hydrogels) and solid (4/34) forms were less frequently used in the studies analyzed. Most studies employed pure CUR (25/34; ~73%), while CUR complexes (2/34—patent-protected formulation), turmeric (4/34—turmeric oil and rhizomes/roots), and curcumin derivatives (3/34) were also reported among the included studies (Figure 2D).

Regarding the types of studies included in this scoping review, 31 out of 34 were preclinical (~91%; Figure 2E), while only 3 were clinical studies investigating the therapeutic effects of CUR nano-based formulations (RCT). In preclinical studies, arthritis was induced through eight different methods (Figure 2F). CFA-induced arthritis was the most common (8/31; ~26%), followed by trauma-induced and collagen-induced models, both with equal representation (7/31). The Gouty and MIA models were each used in three studies (3/31), while the formalin, antigen-induced, and CFA–collagen combination models were each reported in a single study. As for the animal models used in the preclinical studies (Figure 2G), four different species were employed: rats (19/31; ~61%), mice (9/31), rabbits (2/31), and chicks (1/31). Regarding the route of administration of the CUR–nanocarrier complex (Figure 2H), the oral route was the most used in the studies (12/34; ~35%), followed by intra-articular administration (11/34). Topical (7/34), intravenous (3/34), and intraperitoneal (1/34) routes were also employed. Lastly, only 11 performed biosafety assessments (Figure 2I).

The Sankey diagram offers a comprehensive visual representation of the interconnections among key elements identified in the included studies (Figure 3). This analytical tool illustrates the distribution of CUR-based formulations across various experimental models and application contexts, highlighting the frequency of each combination and areas where evidence is limited. By demonstrating the flow between variables, this diagram provides an integrated perspective on the current research landscape regarding nanostructured CUR in arthritis treatment.

The Sankey diagram shows that organic nanoparticle-based systems, particularly nanoparticles and nanoemulsions, are the predominant choices in research and are closely associated with preclinical rodent models, especially those induced by CFA or collagen. An emphasis on liquid formulations was noted, with particular attention given to intra-articular and oral routes of administration, representing the most investigated strategies to enhance CUR bioavailability at sites of inflammation. Conversely, the diagram underscores gaps in existing literature. Despite their promising potential for human use, a few studies focus on clinical models, solid or semisolid formulations (such as hydrogels or films) intended for topical administration, which would suit joint application. Furthermore, there is a lack of safety assessments essential for clinical translation.

Lastly, regarding the quality assessment, in the analysis conducted across 10 domains (Appendix A), most studies displayed an uncertain risk of bias (Figure 4). The bias risk evaluation conducted with the SYRCLE tool showed that all included animal studies (*n* = 31) demonstrated unclear practices regarding allocation sequence generation, baseline characteristics, and allocation concealment. This lack of clarity highlights significant transparency issues in the randomization protocols employed across these studies. The majority were classified as having an uncertain risk of bias, except in the domains of selective outcome reporting and other biases, such as conflicts of interest related to grants or funding. In these instances, 100% (31/31) of the studies were categorized as having a low risk of bias. Furthermore, concerning the blinding domain related to bias detection, approximately 75% of the studies (24/31) were deemed to have an uncertain risk of bias. On the other hand, studies showed an unclear risk of bias for incomplete outcome data, while most demonstrated a low risk of bias in the selective outcome reporting domain. The quality of the clinical studies was assessed using the efficacy outcome because no side effects or safety outcomes were evaluated (Figure 5). The evaluation by the RoB 2 tool revealed that two of three studies demonstrated low risks of bias, while just one was classified as high risk.

## 4. Discussion

A comprehensive narrative synthesis of the findings from the selected studies will be organized according to key aspects: characteristics of the formulations, sources of CUR, the experimental design and schedule, and the primary outcomes observed in the studies.

### 4.1. Formulation

This scoping review is organized according to the different categories of nanocarriers, offering an in-depth analysis of their applications and the advantages inherent to each type. The review delves into the varied compositions of nanomaterials while also discussing their physical and chemical properties in detail.

#### 4.1.1. Nanoparticles

Nanoparticles are structures that have all external dimensions within the nanoscale, displaying minimal variation between their longest and shortest axes [60]. They exhibit a variety of shapes and sizes and have emerged to enhance the bioavailability of therapeutic agents, such as CUR. Studies demonstrated that the encapsulation of CUR in nanoparticles improves its chemical stability, prevents degradation due to enzymatic activity and pH variation, and increases its circulation throughout the body [19,61,62]. Moreover, according to Dewangan and coworkers (2017) [22], the greater release of nanoparticles occurs due to the reduction in particle size, generating an increase in the specific surface area and, consequently, increasing the dissolution of CUR from the particles.

Based on the reviewed articles, most nanoparticles were spherically shaped, and the size ranged between 20 [55] and 300 nm [47]. The polydispersity index (PDI) varied between 0.19 [22] and 0.30 [44], indicating a proximity to a monodisperse size distribution pattern. The zeta potential covered a range from −44.5 [52] to +20.5 mV [21], and the encapsulation efficiency was exclusively mentioned by Niazvand and colleagues (2017) with a value of 97.00 ± 0.45% [37].

The nanoparticles exhibited diverse compositions, encompassing both inorganic and organic constituents. Among inorganic formulations reviewed were zinc oxide nanoparticles [59], which showed an irregular shape; silver nanoparticles (AgNPs) [48]; and gold nanoparticles (AuNPs) [55]. Campos and collaborators (2017) [36] modified AuNPs with poly(allylamine hydrochloride) (AuNP-PAH-CUR). Within organic formulations, the polymer poly lactic-co-glycolic acid (PLGA) was used to prepare nanoparticles [37,46]. Javed and collaborators (2024) [52] designed pure CUR nanoparticles, whereas Zhang and coworkers (2016) [35] formulated chitosan nanoparticles containing CUR. Lastly, hyaluronic acid (HA) loaded with nanoparticles (with dexamethasone and CUR) and tannic acid (TA) and ZnCl_2_ nanostructures were also prepared [47,58].

Additionally, in vitro drug release tests were conducted in a few studies. Dewangan and colleagues (2017) explored the use of polymeric carboxymethylcellulose acetate butyrate (CMCAB) nanoparticles, which are advantageous, since amorphous CMCAB is effective in trapping CUR, forming a solid dispersion and enhancing the drug release [22]. Their findings revealed that nearly 82% of CUR was released from these nanoparticles after 5 h in PBS (phosphate-buffered saline) buffer at pH 6.8 and 37 °C, indicating an enhancement in the release profile compared to the free drug (37%). Another study showed the release profile of CUR from glycyrrhizic acid nanocomplexes (GA/CUR), using the dialysis diffusion method in a PBS solution at pH 7.4. Their results demonstrated that most of the CUR was released within 12 h (95.6%) [44]. Wang and coworkers (2023) prepared CUR-functionalized chitosan–catechol nanoparticles [21]. The drug release behavior was assessed using PBS buffer at pH levels 7.4 and 5.5, both with and without 0.1 M H_2_O_2_. The findings indicated that CUR was released faster at pH 5.5 or after adding 0.1 M H_2_O_2_. Acid-activatable polymeric CUR nanostructures released over 95% of CUR within 7 days at acidic pH, whereas at neutral conditions, the release was slower [42]. Lastly, Hu and colleagues (2023) developed the Hyaluronic Acid-FmoccFF peptide to prepare CUR nanoparticles. According to the test, placed in dialysis bags in PBS (pH 7.4) containing Tween 80 at 37 °C, CUR exhibited a slow release from the nanoparticles [49].

#### 4.1.2. Nanoemulsions

Nanoemulsions are finely dispersed systems consisting of two immiscible liquids, with one phase dispersed as nanoscale droplets within the other, typically stabilized by surfactants [63]. These formulations can be administered via various routes, including topical, oral, and intravenous applications [64]. Furthermore, nanoemulsions can be transformed into a range of formulations, such as gels, creams, and liquids, depending on the desired application and delivery mechanism [65]. These nanostructures enhance the delivery of CUR by improving its solubilization and protection against enzymatic hydrolysis or chemical degradation [66]. Furthermore, as a topical carrier system, nanoemulsion formulations demonstrate great potential due to their low skin irritation, enhanced permeation, and high drug-loading capacity [67,68].

The sizes of nanoemulsions ranged from less than 50 [33] to 1447 nm [50], while the PDI varied from 0.173 to 0.924. Zheng and collaborators (2015) developed nanoemulsions containing CUR using Solutol-HS 15 and soybean oil and studied CUR’s release profile [34]. After 2 h, less than 25% of the CUR had been released, with approximately 30% of the compound released over 24 h. Furthermore, more than 70% of the drug remained associated after 24 h, indicating effective association in the oily core of nanoemulsion. Jeengar and coworkers (2016) explored nanoemulsions containing CUR based on emu oil as the oil phase and Cremophor RH 40 and Labrafil M2125CS as the surfactant and co-surfactant, respectively [25]. Emu oil is similar to skin composition, thereby facilitating CUR’s penetration. Nano-based semisolids were also obtained by converting the nanostructures into semisolids for topical application [33,50].

#### 4.1.3. Other CUR-Loading Nanocarriers

Micelles are nanoscale aggregates that form when surfactant molecules self-assemble in liquid environments at concentrations exceeding the critical micelle concentration. These structures exhibit amphipathic characteristics, consisting of hydrophilic head groups and long hydrophobic tails. This molecular arrangement facilitates the effective encapsulation of hydrophobic compounds within their internal core, thereby shielding the hydrophobic segments from the aqueous environment. In aqueous media, micelles display a hydrophobic core that can incorporate lipophilic substances, significantly enhancing their solubility and bioavailability while simultaneously reducing the potential for degradation and loss of the active compounds. The amphipathic nature of micelles makes them particularly advantageous in pharmaceutical formulations, especially for the delivery of poorly water-soluble drugs, enabling improved therapeutic efficacy and patient outcomes [69]. Among the reviewed articles, CUR nanomicelles were investigated in two RCTs [40,43] and a preclinical study [51]. In this sense, Wang and collaborators (2024) developed ROS-responsive folate-modified micelles using Soluplus, d-Alpha Tocopheryl Polyethylene Glycol, and PEG5000 [51]. These micelles, averaging 90 nm in diameter, were prepared by thin-film dispersion and exhibited a spherical morphology, a negative zeta potential (−6.77 ± 0.67 mV), and a high encapsulation efficiency of 88.66 ± 1.12%. In vitro drug release tests were performed in three different conditions: (A) saline, (B) saline supplemented with 10% fetal bovine serum, and (C) PBS containing 1 mM H_2_O_2_ at pH 7.4. The findings indicated that the CUR release rate was approximately 50% in conditions A and B after 48 h, while in PBS with H_2_O_2_, the release rate increased to 70%.

Another type of nanocarrier is liposomes, which are vesicles composed of phospholipid bilayers that carry both hydrophobic and hydrophilic drugs and can facilitate in vivo delivery of CUR [70]. These vesicles can vary in structure, from unilamellar to multilamellar forms [71]. The lipidic structure of liposomes provides an internal shelter that protects encapsulated drugs from degradation or modification, thereby enhancing their circulation lifetime and accumulation at targeted sites [72]. Among the evaluated studies, liposomes encapsulating dimethyl CUR (Lipo-DiMC) using soybean phosphatidylcholine and cholesterol [38], and BDMC-conjugated vitamin E TPGS liposomes using ethanol, lecithin, cholesterol, vitamin E TPGS, and ethanol containing BDMC were prepared [20]. Liposomes studied by Wang and collaborators (2021) exhibited a PDI value of 0.35 (~±0.016), negative zeta potential (−38.21 ± 0.29 mV), drug loading capacity of 4.84%, high encapsulation efficiency (96.98 ± 0.17%), and spherical morphology [20]. Furthermore, in vitro drug release tests revealed that in acidic (pH 1.2) and neutral (pH 7) conditions, 78 and 68% of BDMC were released from the liposomes within 48 h, respectively.

Regarding lipid composition, liposomes and transferosomes have morphological similarities. However, transferosomes represent a distinct class of deformable vesicles that can travel through pores smaller than their size. They typically exhibit a nanoscale vesicular size (less than 300 nm) and enhanced elasticity, generally 5 to 8 times greater than that of conventional liposomes [73,74,75]. Due to its good penetrating power and flexibility, this carrier system is used for transdermal delivery of drugs of diverse molecular weights. Furthermore, transferosomes consist of deformable and flexible lipid-based supramolecular aggregates, characterized by at least one internal aqueous compartment surrounded by an adapted lipid bilayer [76,77]. The amphipathic characteristics of this nanostructure facilitate the encapsulation of drugs with varying solubilities. Additionally, the formulation incorporates natural phospholipids and edge activators, which contribute to its biocompatibility, biodegradability, and non-toxicity, making it an ideal candidate for therapeutic applications [78,79]. In this regard, Sana and collaborators (2021) investigated CUR-loaded transferosome gel, formulated with phospholipon 90 G [23]. Based on tests, the selected formulation was spherical, measuring approximately 219 nm, and exhibited a 0.20 PDI, negative zeta potential (−17.1 mV), and high encapsulation efficiency (76 ± 2%). In vitro drug release tests were executed at pH levels of 5.5 and 7.4, encompassing both gel and non-gel formulations. The results indicated that both nano-based systems presented an initial slight burst release of CUR, followed by a sustained release over an extended period, in contrast to plain CUR.

Nanocapsules constitute a reservoir system comprising an aqueous or oily core encapsulated by a polymeric or lipidic shell [80,81]. This configuration effectively protects the surrounding material from adverse environmental conditions while facilitating its controlled and targeted release [82]. Nanocapsules offer advantages over liposomes and nanoparticles, particularly in their high drug-loading capacity without the risk of premature leakage. Moreover, they exhibit enhanced stability during storage compared to other nanocarriers, along with the versatility for surface modification and toxicity moderation of highly toxic drug molecules [83]. The study conducted by Sun and collaborators (2024) [54] developed spherically shaped chondroitin sulfate-modified tragacanth gum–gelatin nanocapsules loaded with CUR nanocrystals. These nanocarriers exhibited an average size of 80.80 ± 16.47 nm, a PDI of 0.337, a negative zeta potential of −35.87 ± 5.90 mV, and a drug loading capacity of 54.18 ± 5.17%. In vitro drug release tests indicated that 90% of CUR was released from the nano-based formulation within 12 h, in the presence of the matrix metalloproteinase-2, suggesting that the formulation could effectively release CUR in the inflamed joints of RA patients, where such a marker is highly expressed.

Microemulsions represent a specialized category of nanocarriers characterized by their clear and transparent appearance. This optical clarity is achieved through the precise formulation, involving a balanced ratio of oil, water, and surfactants, which facilitates the formation of thermodynamically stable dispersions [84]. Due to their small droplet size, optical isotropy, and thermodynamic stability, they are an excellent vehicle for drug delivery in topical applications. Microemulsions can deliver poorly water-soluble drugs, such as CUR, effectively to the epidermis and deeper layers of the skin [85]. This system facilitates sustained release, minimizes adverse effects, and enhances drug solubility, protecting active compounds against degradation, hydrolysis, and oxidation [86]. Consequently, the incorporation of CUR into microemulsions is beneficial for transdermal and topical delivery [87], enhancing solubility [88] and oral bioavailability [89], and demonstrating anti-inflammatory effects in the context of RA [45]. Zhang and collaborators (2022) developed a CUR-laden amphiphilic chitosan microemulsion [45]. The formulation consisted of chitosan combined with oleic acid and CUR. The resulting average particle size was approximately 12.31 nm, with a PDI of about 0.054.

Nanosponges are another type of nanocarrier that falls within the nanoscale. They are mesh-like structures with a 2D network that can encapsulate various substances and drug molecules. Their composition includes long-chain polyesters dissolved in a solution, along with crosslinkers that interconnect the polymer components [90]. This carrier presents a spherical colloidal form and a well-recognized potential for hydrophilic and lipophilic solubilization enhancement. Furthermore, nanosponges contribute to the increased bioavailability of components through a prolonged release mechanism [91]. Their internal hydrophobic structure, combined with external hydrophilic branches, confers an amphiphilic nature to this nanostructure, enabling the transportation of both hydrophilic and hydrophobic therapeutic agents. Xu and collaborators (2024) developed spherical CUR-loaded biomimetic nanosponges that presented a nanometric size, negative zeta potential (−14.89 mV), and 5.6% drug loading capacity, presenting gradual release of CUR over time [56].

Solid lipid nanoparticles (SLNs) and nanostructured lipid carriers (NLCs) represent two prominent types of lipid-based nanosystems [92]. In comparison to liposomes, they provide drug stability and prolonged release characteristics while also being safer than polymeric carriers due to the absence of organic solvents in their production process [93]. SLNs consist of solid lipids dispersed in an aqueous phase and surfactants that improve stability [94]. Among their many advantages over other systems, SLNs provide excellent physical stability, demonstrate no toxicity associated with the lipid carrier system, and exhibit biodegradability of the lipids used [95]. Among the reviewed articles, Arora and coworkers (2014) prepared SLNs containing CUR that were composed of Polysorbate 80, soy lecithin, and lipid (Compritol 888 ATO) [24]. The size was approximately 134.6 ± 15.4 nm, according to Mastersizer analysis, and between 40 and 120 nm, as observed in transmission electron microscopy. The encapsulation efficiency was about 81.92 ± 2.91%, and the nanocarrier exhibited a spherical morphology. Nevertheless, SLNs have certain limitations, including a tendency to undergo gelation and having an inherently low drug incorporation rate due to their crystalline structure [93,94]. To address these challenges, NLCs have been developed [96]. They are modified SLNs composed of solid and liquid lipid phases at ambient temperature and enhance stability and drug loading capacity while mitigating the risk of drug release during storage [97]. Shinde and colleagues (2020) developed NLCs composed of cetyl palmitate, Labrafac PG, Captex 200, Tween 80, and Labrasol, through hot homogenization and subsequent melt ultrasonication [41]. The resulting NLCs exhibited a negative zeta potential of −21.67 mV, a high encapsulation efficiency of 72.15%, and an average size of 165.12 nm. They also conducted an in vitro drug release test by dispersing CUR-NLC in a phosphate buffer (pH 7.4) at 37 °C. The data demonstrated that the cumulative release of CUR varied between 87.73% for formulation CF-9 and 99.27% for CF-1 at the conclusion of the 84 h period. The optimized formulation, identified as CF-10, exhibited a release rate of 94.32%, closely aligning with the predicted release of 96% at the same time point. These data reinforce the reliability of the checkpoint analysis utilized in this assessment.

Niosomes serve as water-soluble carriers that, when subjected to brief agitation in a hot aqueous medium, transform into a niosomal dispersion. This dehydrated form is referred to as a proniosome. Proniosomes consist of microscopic lamellar structures incorporating a non-ionic surfactant and cholesterol, subsequently hydrated in aqueous solutions. Similar to liposomes, they are made of a bilayer formed by non-ionic surface-active agents [98] that enhance efficacy, reduce or eliminate adverse effects, and improve therapeutic actions of drugs. They are employed to avoid gastrointestinal incompatibility, pre-systemic metabolism, and undesirable side effects associated with oral delivery. Furthermore, proniosomes sustain therapeutic drug levels over extended periods, reduce the frequency of administration, and enhance patient compliance [99,100]. Kumar and Rai (2012) prepared four formulations of a proniosomal gel of CUR, composed of surfactant and cholesterol [32]. The particles are spherical and exhibit release rates of approximately 59% and 85%. The formulation presented a high CUR association and stability.

Lastly, exosomes are vesicles ranging in diameter from 30 to 150 nm. They exhibit excellent biocompatibility, low immunogenicity, and intrinsic targeting properties, all of which contribute to their efficacy in drug delivery applications [101]. These vesicles are bioproducts derived from various cell lines, including immune, cancer, stem cells, and bodily fluids [102,103]. Their structure features a lipid bilayer membrane containing embedded tetraspanins, glycoproteins, and signaling receptors, along with DNA and microRNA [104]. Due to its properties, a study demonstrated that exosome-encapsulated CUR enhanced its solubility, stability, and bioavailability [105]. Beyond their role in intercellular communication, exosomes can transport various bioactive molecules, including proteins, lipids, and nucleic acids [106]. In comparison to conventional synthetic nanoparticles, this nanostructure exhibits superior targeting capabilities, improved cellular uptake, and more controlled drug release properties [107]. Zhang and colleagues (2024) designed CUR-loaded exosomes derived from primary bone marrow-derived mesenchymal stem cells modified with HA and polyethylene glycol [57]. The carriers had a spherical morphology, with a diameter of 131.58 nm, a negative zeta potential (−14.68 mV), and a low encapsulation efficiency of 29.78%. Regarding the in vitro release profile, the NLC developed by Shinde and colleagues (2020) was the only one demonstrating a full release in 108 h [41]. In this context, among the articles that investigated drug release, some specifically compared the release profiles of free CUR with those of a nano-based formulation. The release of BDMC from liposomes was approximately 3.9-fold higher at pH 1.2 (78%) and 3.7-fold higher at pH 7.0 (68%) compared to the release profile of the free suspension within 48 h of the experiment [20]. Conversely, liposome-containing BDMC demonstrated a more sustained release over time. Similarly, in another study, more than 90% of CUR was released from free suspensions within the first 30 min [34], while less than 25% and approximately 30% of the drug were released from NLC at 2 and 24 h, respectively. Additionally, more than 70% of the drug remained encapsulated in the nanoemulsion after 24 h. A slow and controlled release profile from nanocarriers was also observed by Sun et al. (2024) [54]. Comparably, CUR nanoparticles at pH 6.8 (simulating small intestine conditions) showed an initial rapid release in the first hour, followed by a controlled release over five hours, totaling approximately 82% of drug release, which is substantially higher than the 37% observed for the free drug [22]. This initial burst release may be attributed to the free CUR adsorbed on the surface of the nanoparticles or loosely associated near the surface. In contrast, the lower release at acidic pH (1.2) likely reflects the insolubility of the polymeric matrix under such conditions. Although these findings collectively suggest that CUR nanoencapsulation may prolong drug availability and contribute to sustained therapeutic levels, direct comparisons between release behaviors should be interpreted with caution. The observed differences are strongly influenced by methodological variables such as the presence or absence of a dialysis membrane, the use of surfactants, and the physicochemical composition of the release medium. These variations, which are often not standardized across studies, make it difficult to establish robust comparative conclusions. This lack of methodological harmonization represents a broader challenge in the field of nanocarrier-based drug delivery and should be considered a limitation of current in vitro release assessments.

Collectively, this topic covered the general characterization of CUR-loaded nanocarriers. The findings indicate that microemulsions exhibited the lowest PDI value (0.054) [21,45], which is favorable and suggests a stronger tendency towards monodispersity. On the other hand, nanoparticles demonstrated the highest zeta potential (+20.5 mV) [21] and achieved an encapsulation efficiency of approximately 97% [37]. Consequently, while nanoparticles may not exhibit PDI values as low as one might prefer, they remain the most widely used model due to their promising characteristics. Furthermore, the in vitro release results demonstrated variability across different pH levels. Liposomes showed enhanced release profiles at acidic pH, whereas nanoparticles performed better at a pH closer to neutrality, when compared to free CUR. This difference can be attributed to the insolubility of the polymer at more acidic pH levels. However, both nanoformulations exhibited improvements in drug release compared to free CUR. Consequently, despite the observed variations in PDI profiles and in vitro release behaviors, each formulation offers unique and specific advantages. The specific therapeutic objectives should determine the selection of the most appropriate system.

### 4.2. Curcumin Source

Despite its pharmacological properties, CUR exhibits several limitations, including poor aqueous solubility, instability in physiological conditions, high metabolic turnover, rapid systemic clearance, diminished gastrointestinal absorption, and limited bioavailability [19,61]. The clinical application of polyphenols has been restricted by various factors, highlighting the necessity for developing nanoformulations to enhance their therapeutic efficacy. While many studies predominantly explored pure CUR, alternative formulations have also been studied. These include Dimethyl curcumin, the CUR C3-complex, a combination of 95% CUR with 5% methoxycurcumin and bis-methoxycurcumin, BDMC, turmeric oil, as well as both dried turmeric rhizomes and fresh roots of *Curcuma longa*.

CUR is the main curcuminoid in turmeric (*Curcuma longa*), with relevance in herbal medicine due to its several properties, such as anticancer, anti-inflammatory, antiviral, and anti-arthritic effects [17,108,109]. Dimethylcurcumin (DiMC) (C_23_H_24_O_6_) is one of the synthetic analogues of CUR (C_21_H_20_O_6_), structurally derived from CUR with minor modifications, which is the methylation of two hydroxyl groups [110,111,112]. Regarding CUR, DiMC showed greatly improved metabolic stability and solubility. Moreover, BDMC, a constituent of turmeric, has been widely studied due to its therapeutic effects, including antioxidant, anticancer, antidiabetic, and anti-inflammatory properties [20]. Another form of this compound is the CUR C3-complex, a purified mixture of CUR, BDMC, and demethoxycurcumin. The turmeric oil was mixed with Tween^®^ 80 [50], while turmeric rhizomes in dry form were associated with AgNPs [48]. Lastly, fresh roots were used to prepare a crude extract of *Curcuma longa* [55].

Even variations of CUR do not address the physicochemical limitations that encapsulation can effectively overcome. For example, BDMC presents similar challenges in clinical applications, exhibiting poor solubility and bioavailability like CUR [20]. However, a study conducted by Wang and collaborators (2021) highlighted that the oral bioavailability of liposomal BDMC was approximately 10 times higher than that of free BDMC, as demonstrated by cumulative release rates both in vivo and in vitro [20]. This indicates an improvement in the sustained effectiveness of BDMC in its encapsulated form compared to its free counterpart. Furthermore, CUR is a crystalline compound that typically requires a considerable amount of energy to dissolve in water [22], meaning that its low solubility presents challenges for the absorption of this drug. In this context, nanoformulations offer a promising solution to overcome this challenge. For instance, nanoemulsions developed by Okpalaku and colleagues (2023) demonstrated enhanced solubilization of lipophilic compounds derived from plant oils [22], such as Turmeric oil, due to their high surface area [113].

Additionally, nanoemulgels, which are essentially nanoemulsions encapsulated within a hydrogel matrix, demonstrate significant potential for enhancing CUR solubility. This nanoformulation not only mitigates enzymatic degradation and hydrolysis of the active compound but also exhibits an improved spreading coefficient when compared to certain commercially available topical delivery systems [114,115]. Moreover, CUR nanoparticles demonstrated a favorable structure, as the amorphous nature of the polymer effectively encapsulates the drug in its amorphous form, resulting in solid dispersion and enhanced drug release [22]. The synergistic impact of increased surface area achieved through decreased particle size, along with the creation of amorphous solid dispersions, plays a vital role in enhancing the release kinetics of nanoparticle-based drug formulations. This strategy results in a significant improvement in bioavailability compared to the poorly soluble crystalline forms of the drug. Thus, the encapsulation of curcumin (CUR) and its derivatives within nanocarriers is critical for optimizing release profiles and augmenting both the pharmacological efficacy and solubility of these compounds. Furthermore, some studies highlighted that encapsulation of CUR in nanosystems can increase its solubility [37,44]. The CUR-loaded PLGA nanoparticles demonstrated excellent dispersion in aqueous medium, showing no signs of aggregation, whereas free CUR displayed limited solubility in water [37]. Similarly, GA/CUR nanocomplexes successfully increased the solubility of CUR, achieving near-complete release [44]. Briefly, it is essential to acknowledge that both CUR and its alternatives present remarkable pharmacological effects. However, they also face several limitations, including low solubility, instability, and bioavailability, which considerably hinder their clinical applicability in treating arthritis. Various CUR formulations have been explored, ranging from the pure form to synthetic derivatives such as DiMC, conventional forms like turmeric oil and fresh root extracts, and commercially available formulas such as C3-complex. Nevertheless, such formulations do not fully overcome the obstacles associated with physicochemical properties. In this regard, nanocarriers emerge as a promising solution, enhancing solubility, safeguarding the drug from degradation, and extending its release and bioavailability.

Lastly, in the reviewed studies, CUR emerged as the predominant compound utilized; however, other preparations like BDMC, turmeric oil, fresh or dried rhizomes, and various commercial C3-complex formulations were also highlighted. While these alternatives share specific anti-inflammatory and antioxidant properties, they should not be considered pharmacologically interchangeable. For example, BDMC demonstrates enhanced metabolic stability but exhibits lower anti-inflammatory efficacy relative to CUR [20]. Additionally, turmeric extracts can present a heterogeneous composition due to varying concentrations of curcuminoids and volatile oils, leading to discrepancies in bioactivity [17]. Most original studies did not perform direct comparative analyses between these forms, often lacking comprehensive physicochemical characterization. Consequently, although these compounds are classified together for the sake of evidence mapping, they must be regarded as distinct entities in pharmacological evaluations. This underscores the necessity for caution when comparing outcomes across different studies and highlights the critical importance of standardized characterization and reporting of these compounds in future research.

### 4.3. Experimental Design of Preclinical and Clinical Studies

This section is structured into two main categories: preclinical and clinical investigations. It provides an in-depth analysis of the key animal models used to decode the mechanisms of arthritis pathogenesis, alongside a detailed examination of the various methodologies employed in this field of research.

#### 4.3.1. Preclinical Studies

Animal models employed in arthritis research can effectively mimic the pathophysiological conditions seen in humans, thus enabling the exploration and validation of novel therapeutic pathways and molecular targets. A variety of species can be utilized to capture the disease’s multifaceted clinical manifestations, with each model providing insights into specific pathological mechanisms. This diversity allows for a more nuanced investigation of the underlying processes involved in arthritis, ultimately informing the development of innovative treatment strategies [116] to enhance the understanding of arthritis development. In this review, six different inductions were applied, and the main animal model used was rats. Additionally, the most reported type of arthritis was OA, followed by CFA-induced arthritis and collagen-induced arthritis (CIA).

OA is characterized by severe articular cartilage degradation [117,118]. The development and progression of this disease are related to inflammation, such as synovitis and systemic inflammation. One potential explanation for this phenomenon is the activation of a foreign body response in synovial cells triggered by the degradation of cartilage, which may exacerbate cartilage destruction. Conversely, other research indicates that activated synovial macrophages and the innate immune system play a crucial role in OA progression [118]. Four studies that induced OA used the medial meniscus destabilization (MMD) method. Campos and collaborators (2017) used mice for the technique [36], Lin and colleagues (2023) used rats to perform surgical fixation of the medial meniscus [47], Xu and coworker (2024) performed surgical transection of the medial meniscotibial ligament in the right hind limb of mice [56], while rabbits were used in the study of Pérez-Expósito and collaborators (2024) to perform a unilateral section of the anterior cruciate ligament [58]. The other studies performed anterior cruciate ligament and partial meniscus transection in rats [21,55], osteochondral defect in the lateral condyle of the right femur in chicks [55], and surgical section of the medial meniscotibial ligament in the right hind limb in mice [35]. Another OA model can be performed using monoiodoacetic acid or mono-iodoacetate (MIA), which is frequently utilized as an agent to induce arthritis in rat models, primarily for the investigation of arthritis-related nociception [119,120]. Kang and colleagues (2020) performed disease induction in mice via the patellar tendon lesion [42], while MIA injected into the knee joints in rats was another approach to OA simulation [37,46].

A very common type of induction is achieved by using CFA, a suspension composed of desiccated and heat-inactivated mycobacteria, specifically, *M. tuberculosis* and *M. butyricum*, dispersed in paraffin oil and mannide monooleate. This adjuvant is widely used in experimental settings to induce autoimmune disorders in animal models. Its effectiveness is attributed to its ability to prolong the presence of injected autoantigens, potentiating the delivery of these antigens to the immune system and stimulating the innate immune response [121]. The selected studies applied CFA-induced arthritis in mice [23,45] or rats [22,24,25,32,33,34,48]. In an adapted model of CFA, RA was induced in rabbits through the administration of a CFA and a collagen type-II emulsion [59].

CIA is another commonly used model for RA due to its ability to replicate immunological and pathological characteristics observed in human RA, such as symmetric joint involvement, synovitis, and destruction of cartilage and bone [122,123]. For example, the murine model is similar in pathogenesis to human RA and widely used due to its speed and low cost [124]. Furthermore, the CIA model is recognized for its stability and is considered an ideal framework for investigating the pathogenesis of RA, as well as for screening potential therapeutic agents for its treatment. The CIA mouse model has disadvantages due to the variable disease pattern and smaller joint size. On the other hand, rat models of arthritis offer much larger specimen sizes and more reproducible distribution and extent of inflammatory changes in CIA joints [125]. Nonetheless, mice are less expensive than rats [126]. Therefore, a CIA mouse model is appropriate for the initial screening of anti-RA drugs, unless the experiment requires a different specimen size or other constraints that preclude the use of mice. For the CIA model, both females and males can be used [127,128,129]. However, studies report that in mice, CIA tends to be more severe in males than in females [128,130]. From the reviewed articles, six induced arthritis by CIA. Three used a rat model [38,49,51], while three were performed on mice [44,54,57].

An additional category of arthritis is gout, which represents the most prevalent form of inflammatory arthritis, primarily resulting from hyperuricemia. This condition is characterized by elevated levels of uric acid, which leads to its deposition in crystal form, known as monosodium urate (MSU) crystals [131], within periarticular [132] and intra-articular tissues [133]. These crystals activate components of the innate immune system, thereby triggering an intense local inflammatory response [134] that initiates the clinical manifestations of gout [131]. In the reviewed studies, Javed and collaborators (2024) conducted experiments by injecting MSU crystals into the right ankle joint, as well as intraperitoneally [52], while another study administered MSU through the intraperitoneal route in rats [20] or in the ankle of mice [39,54].

In another approach, antigen-induced monoarthritis is established through local injection of antigens into the joint of a hyperimmunized animal. It can be induced in various species, including mice, rats, and guinea pigs, if the requisite immunity to a specific antigen can be achieved. Commonly used antigens include ovalbumin, bovine serum albumin, and fibrin. This form of arthritis is localized to the injected joint, allowing for a comparative analysis of biochemical and structural changes in the arthritic joint versus the healthy contralateral joint. The severity of the condition can be modulated through the dosage of the locally injected antigen, and the arthritis demonstrates a well-defined onset time [135]. Among the reviewed studies, only Shinde and collaborators (2020) used an antigen-induced monoarthritis model in rats [41].

Lastly, formaldehyde can be used to trigger arthritis due to its mechanism of denaturing proteins at the injection site, which leads to an immune reaction against these degraded materials [136], which was used by Okpalaku and collaborators (2023) [50]. The arthritic process is initially marked by the release of substance P, a neuropeptide that can cause neurogenic inflammation [137]. These chemical mediators play an important role in hyperalgesia by activating nerve endings and pain receptors, resulting in increased sensitivity.

In summary, the animal models outlined here offer essential insights into the pathophysiological mechanisms of human arthritis. The CIA model stands out as the most widely employed system for studying RA due to its capacity to replicate critical features of the human condition. Its advantages include rapid disease onset and cost-effectiveness, rendering it a preferred choice for research investigations. In contrast, the CFA model is commonly used to provoke autoimmune disorders in experimental animals. Furthermore, alternative models such as OA, gout, antigen-induced monoarthritis, and formaldehyde-induced arthritis serve as valuable options for preclinical assessments in arthritis research.

#### 4.3.2. Clinical Studies

Three clinical studies were reviewed. Despite the difference in protocols and doses, two studies used the same commercial formula (CUR C3-complex) [40,53], composed of a purified mixture of CUR, BDMC, and demethoxycurcumin [138]. In contrast, Ahmadi and collaborators (2020) used CUR in nanomicelles [43].

Lustberg and colleagues (2024) conducted a multi-institutional randomized, double-blind, placebo-controlled pilot study to investigate the efficacy of CUR C3-complex [53]. This article looked at arthralgia induced by aromatase inhibitors, a medicine used to reduce the risk of recurrence and death in postmenopausal women with breast cancer. It involved 42 eligible postmenopausal women diagnosed with estrogen receptor-positive breast cancer who were experiencing this type of arthralgia. The participants were divided into two groups, with 22 randomly assigned 1:1 to the intervention group and 20 to the placebo group. However, eight women did not complete the 3-month study. Ahmadi and coworkers (2020) organized a 16-week randomized, double-blind, placebo-controlled trial [43]. The study comprised 24 male patients diagnosed with ankylosing spondylitis through clinical assessments. These individuals were stratified into two groups: a control group and a treatment group, each consisting of 12 participants. Additionally, Javadi et al. (2019) conducted a randomized, double-blind, controlled trial involving 65 eligible patients diagnosed with RA, aged between 20 and 80 years [40]. The participants were randomly assigned to the CUR-loaded nanomicelle (*n* = 30) and the placebo (*n* = 35) group. Nonetheless, at the end of the study, clinical assessments were conducted on only 49 patients due to withdrawals from participation or loss to follow-up.

In one of the studies, women were instructed to take one capsule (200 mg/day) containing either nano-based CUR or a placebo twice a day for three months. The placebo capsules were formulated with the same excipients to replicate the CUR C3-complex containing capsules [53]. On the other hand, Ahmadi and colleagues (2020) treated one group with nano–CUR capsules (dose of 80 mg/kg) day-to-day for 4 months, while the control group received placebo capsules [43]. Lastly, CUR-loaded nanomicelles (40 mg) and placebo capsules with wheat flour (500 mg) were administered to the patients three times a day for three months [40].

The nano-based CUR was well-tolerated but did not show significant benefits over placebo in treating arthralgia induced by aromatase inhibitors. Study limitations may have impacted the results, including a small sample size and potential placebo effects influenced by public perception of CUR [53]. In contrast, another study revealed that nano–CUR administration for four months has potential as a novel therapy for ankylosing spondylitis, as it modulated Treg-associated miRNAs and significantly increased Treg populations and anti-inflammatory mediators (FoxP3, TGF-β, IL-10) while suppressing IL-6 in affected patients [43]. The CUR nanomicelle demonstrated a significant reduction in the Disease Activity Score for 28 joints (DAS-28), Tender Joint Count (TJC), and Swollen Joint Count (SJC) in the treatment cohort relative to the placebo cohort [40]. Overall, there were no significant improvements with the CUR nanocarrier over the placebo because the dosage was low. Also, considering that the nanomicelle was indeed bioavailable, it was either insufficient or inefficient in RA. Limitations potentially include a small sample size and a short duration of the study.

Hence, the RCTs indicate that nano-based CUR formulations are well tolerated and exhibit potential anti-inflammatory effects. Nonetheless, further clinical trials are necessary to evaluate their efficacy compared to a placebo group accurately. Methodological limitations, including small sample sizes, short test durations, and the potential influence of placebo effects, currently restrict the real therapeutic effectiveness of these nanocarriers.

### 4.4. Efficacy Outcomes from Preclinical Studies of Nano-Based CUR

In preclinical studies, it is common to utilize various assessments, including behavioral tests, to evaluate the degree of nociception and discomfort in animals. These assessments can also examine related locomotor behavior. Additionally, a range of scores related to inflammation and cardiac damage, and biochemical and molecular markers from the affected tissues or blood samples that indicate systemic inflammation are often employed. The studies investigated the application of several of these methodologies, and to facilitate a more cohesive and objective discussion, the most relevant findings have been summarized.

#### 4.4.1. Protocol and Administration Schedule

The assessment of the reviewed literature indicated the exploration of five distinct administration routes for nanocarriers. Notably, intra-articular administration was reported in 11 studies, making it the most prevalent route. Oral and topical routes were followed, cited in nine and seven articles, respectively. Intravenous administration was discussed in three articles, while the intraperitoneal route was mentioned in one study.

The analysis of various studies indicates a consistent pattern in the administration of nano-based formulations containing CUR, characterized predominantly by repeated applications, independently of the treatment route employed (Table 2). Intra-articular applications were conducted over several days or weeks, except for the study by Xie and colleagues (2024), which used a single application at the lesion site [55]. Similarly, topical formulations adhered to a regimen of repeated applications, varying from a few days to several weeks. On the other hand, administration via the oral route typically involves daily dosing, with treatment regimens commonly lasting approximately two weeks. Lastly, intravenous administration involved repeated injections throughout 2 to 4 weeks, while the intraperitoneal route was utilized with interspersed applications spread across 24 days.

#### 4.4.2. Experimental Approaches for Assessing Pharmacological Outcomes

Recent studies have highlighted the beneficial implications of integrating drugs with nanocarrier systems, showcasing a spectrum of positive effects, including anti-inflammatory, anti-arthritic, anti-gout, antioxidant, therapeutic, and protective properties. The efficacy of these nanocarrier systems is influenced by the diverse compositions of various nanoparticle types, which can significantly impact drug delivery mechanisms and therapeutic outcomes. For instance, some of the selected studies have shown that PLGA nanoparticles are recognized as effective biodegradable polymeric carriers due to their low toxicity, controlled and sustained-release characteristics [37,46], and compatibility with biological tissues and cells [139]. Furthermore, some authors highlight the application of high-molecular-weight HA [49,58], a valuable component in nanoparticle formulations due to its anti-inflammatory properties [140]. Nevertheless, it is noteworthy that, as a natural polymer, HA is susceptible to degradation into its low-molecular-weight form, which can diminish its lubricative abilities and reduce its anti-inflammatory efficacy [141]. This approach highlights the application of CUR in RA treatment in combination with HA, given its biological properties.

Dewangan and colleagues (2017) revealed that CUR nanoparticles exhibited a controlled release profile and enhanced efficacy compared to free drug in most applications [22]. Similarly, another study reported that treatment with acid-activatable curcumin polymer (ACP) micelles produced more pronounced effects than equivalent doses of free CUR, particularly in joints and in pro-inflammatory marker reduction [42]. These findings underscore the potential of nanoparticles to improve therapeutic outcomes in related applications. Additionally, Cur@Zn&TA nanoparticles were able to significantly reduce inflammatory markers, such as COX-2, IL-6, and TNF-α, when compared to free CUR [47]. Moreover, the nanoparticles enhanced CUR antioxidant potential [59] and significantly inhibited the development of OA [21], with superior therapeutic effects since it enabled cartilage-targeted release. Conversely, oral administration of free CUR and topical application of CUR nanoparticles effectively slow the progression of post-traumatic OA [35]. Campos and collaborators (2017) demonstrated that intra-articular application of AuNP-PAH-CUR showed a lower severity of histological lesions [36].

Other nanocarriers that used pure CUR have also demonstrated beneficial effects on arthritis and related conditions. Studies that used topical treatment showed that CUR can bypass first-pass metabolism and contribute to joint synovial membrane [25], improve the solubility and stability of CUR, and did not damage the skin structures [45], and provided controlled skin permeation with significant anti-inflammatory and anti-arthritic effects with a comparable efficacy to indomethacin [32]. Nanomicelles outperformed free CUR by prolonging drug circulation, increasing cartilage protection, and amplifying anti-inflammatory effects [51]. Similarly, rats treated with CUR-NLC gel formulation had greater reductions in joint inflammation compared to those receiving free CUR, benefiting from sustained release and progressive improvement [41].

Furthermore, the CUR-transferosome hydrogel has shown superior therapeutic efficacy over free drug by effectively facilitating CUR delivery through the skin and reducing inflammation, edema, and nociception associated with arthritis [23]. Similarly, another study reported that cumulative transdermal CUR after 24 h from CUR-loaded microemulsions was doubled compared to the free CUR group [45]. This finding indicates that microemulsions play a relevant role in promoting the transdermal process of drugs and can also promote the transdermal absorption of CUR.

Of particular importance, following the oral administration, the levels of inflammatory markers observed in the CUR–nanoemulsion-treated group were found to be more than two-fold and three-fold lower in the synovial fluid and the blood serum, respectively, when compared to those in the suspension group [34]. Comparably, the mRNA expression of pro-inflammatory factors in the ankle joint was more downregulated in the group treated with nano-based CUR, highlighting the potential of nanocarrier technologies in optimizing the therapeutic outcomes for arthritis treatment [54].

The nano-based formulations containing turmeric rhizomes and BDMC have shown potential as a promising anti-gout therapy [20,39]. The dried turmeric rhizomes were also associated with AgNPs and exhibited anti-inflammatory effects, effectively reducing arthritis progression and improving the symptoms of RA [48]. Similarly, Okpalaku and colleagues (2023) investigated the properties of turmeric oil in a nanoemulgel formulation, which initially showed a reduction in paw volume [50]. However, it demonstrated lower efficacy in comparison to other formulations. Furthermore, Lipo-DiMC has shown therapeutic potential in mitigating the progression of CIA in rat models [38]. In another study, CUR-SLNs presented a dose-dependent improvement of various arthritic signals in rats [24]. In addition, CUR-SLN treatment enhanced biochemical markers and preserved the radiological integrity of the joints. Wang and colleagues (2021) compared the efficiency of using nanocarriers when compared to free drugs, demonstrating that the oral bioavailability of BDMC-loaded liposomes is approximately 10 times greater than that of free BDMC, as evidenced by both in vivo and in vitro cumulative release rates [20]. Lastly, AuNPs containing *Curcuma longa* roots extract have promising therapeutic effects in knee osteoarthritis, reinforcing its potential for treatment applications [55].

The most frequently used behavioral tests were thermal hyperalgesia on the hot plate, mechanical hyperalgesia, mechanical allodynia, and locomotion-related aspects. In addition, the assessment included other behaviors, such as motor incoordination, thermal hyperalgesia on a cold plate, joint hyperalgesia, paw licking response, tiredness, stiffness of the limbs, aggressive behavior, reduced appetite, discomfort, hind limb rearing, and tactile sensibility (Von Frey testing).

The use of nanocarriers with CUR effectively alleviated thermal nociception, which was evidenced by an increased reaction time and an enhancement in the paw licking response observed during the hot plate test, indicating a potential antinociceptive effect in some studies [20,22,23,24,25]. Moreover, mechanical hyperalgesia and allodynia were also improved after treatment with nanocarriers of CUR, specifically regarding an increase in the nociceptive threshold [22,24]. Lastly, the treatment with nanocarriers restored the impairment in activity/locomotor behavior of the arthritic animals [22,35,54,59].

Of particular importance, Arora and colleagues (2014) revealed that for the three assessments (thermal hyperalgesia, mechanical hyperalgesia, and mechanical allodynia), free CUR administered at a dosage of 10 mg/kg did not produce any significant therapeutic effect in CFA-induced arthritic rats, in contrast to SLNs and free CUR at a higher dosage of 30 mg/kg [24]. Another study applying the same behavioral assessments demonstrated that CUR nanoparticles exhibited greater efficacy in rapidly alleviating symptoms of thermal hyperalgesia and enhancing mechanical hyperalgesia when compared to free drug [22]. In the study conducted by Sana and colleagues (2021), the hot plate method was employed to assess the antinociceptive effects of CUR-loaded transferosome gel on thermal nociception [23]. The group treated with nano-based formulation demonstrated a reduction in arthritis symptoms. Additionally, CUR-loaded microemulsions restored the paw licking response [45]. The antinociceptive effect of free BDMC did not meet therapeutic requirements, while the liposome-containing BDMC demonstrated effective antinociceptive properties by prolonging latency time in response to heat [20]. Furthermore, administration of high doses of liposome formulation prolonged the nociceptive threshold response compared to the positive control group (ibuprofen). Topical treatment with CUR nanoparticles reduced mechanical sensitivity and enhanced locomotor behavior compared to vehicle-treated mice [35,54]. Therefore, encapsulation of CUR within nanostructures has contributed to improving behavioral assessments.

Regarding the scores, most studies focused on paw-related parameters such as swelling, edema, volume, thickness, and redness. In addition, radiographic assessments and arthritic scores were frequently mentioned in the analysis. Other scores addressing the knee, ankle, joints, bones, erythema, and mobility were also assessed, yielding favorable results. All studies that evaluated paw parameters indicate a considerable decrease or prevention of their increase after treatment with nano-based CUR formulation. Zheng and colleagues (2015) demonstrated that the paw swelling rate for methotrexate and CUR was similar, with both treatments showing reductions when compared to the model group [34]. On the other hand, niosomes containing CUR exhibited a lower mean percentage of paw volume inhibition than indomethacin [32]. Lastly, the administration of AgNPs reduced paw edema and exhibited better anti-inflammatory effects than dexamethasone [48].

Few articles analyzed imaging exams, including radiographs, micro-CT (Micro-computed tomography), and macro-photographs. The samples included joints, bones, cartilage, the articular space of the joint, paws, and soft tissues. All articles evidenced improvements in various arthritic parameters. As evidenced by Jeengar and collaborators (2016), soft tissue swelling, deformity, and erosion in the ankle joints were suppressed in both high- and low-dose groups treated with the hydrogel containing CUR-loaded nanoemulsion (Figure 6B) [25]. Remarkably, this study evaluated specific scores, including the Osteoarthritis Research Society International (OARSI) score, indicating a marked improvement in the arthritic clinical score after the topical administration of CUR-loading nanoemulsions gel formulations (Figure 7A,B) [25].

All studies that compared free CUR with CUR in nanocarriers and assessed scores have consistently shown that both formulations exhibit anti-inflammatory properties. Nevertheless, findings suggest that nanotechnology frequently produces superior outcomes. In the study conducted by Arora and colleagues (2014), the results indicate that Naproxen, free CUR (30 mg/kg), and CUR-SLNs (10 and 30 mg/kg) significantly improved the mobility score, effectively prevented the increase in paw volume, and alleviated joint stiffness in CFA-injected rats [24]. Furthermore, CUR-SLNs (30 mg/kg) produced effects comparable to those of naproxen (25 mg/kg) in paw volume and joint stiffness. In comparison, free CUR (10 mg/kg) did not significantly affect CFA-induced arthritic rats in the mobility score. Similarly, other work reported a faster recovery in rat mobility on regular treatment with nanoparticles compared to CUR, and both formulations were found to be highly effective in addressing joint stiffness [22]. In addition to that, some studies indicated that both free CUR and CUR incorporated in nanocarriers effectively prevented the increase or reduced paw swelling, with the nanocarrier formulation exhibiting enhanced therapeutic efficacy [22,23,45,51]. This is illustrated in Figure 8A,C, which reports the effectiveness of various CUR formulations in mitigating paw thickness [51]. Specifically, panel A reports that the CUR-loaded micelles formulation was the most effective in restoring the hind paw, and panel C reinforces that the reduction in paw thickness was greater for the nano-based formulation compared to the other treatments.

Data derived from radiological evaluations suggest that the pharmacological efficacy of CUR is enhanced when delivered orally through nanoparticle formulations or topically via transferosome carriers [22,23]. Moreover, the study of Sun and collaborators (2024) evaluated the effects of free CUR and CUR nanocapsules on both RA and gouty arthritis [54]. While free CUR moderately ameliorates symptoms such as redness, swelling, and articular deformity in RA, CUR-loaded nanocapsules exhibited greater therapeutic effects than free CUR and inhibited the progression of gouty arthritis. Furthermore, Zhang and coworkers (2022) scanned via micro-CT the hind paws of the mice, revealing that the gel of CUR microemulsions was more effective in alleviating the swelling and erosion in the joints than the formulation containing free CUR [45]. Additional analysis of the OARSI score revealed that both free CUR and its nano-based formulations effectively reduced the score, with nanoparticles showing greater reduction [21,35,47]. Lastly, some studies examined the effects on the animals’ knees [21,41]. A marked reduction in cartilage degeneration and increased articular space width following the administration of CUR and CUR-loaded nanoparticles was observed. Remarkably, the nanoparticles exhibited superior inhibitory effects compared to the free CUR formulation [21]. After 21 days, CUR-NLC gel produced a significant reduction in knee inflammation compared to treatment with free drug, with NLC lowering to 34% and CUR to 85%, approximately [41].

Most preclinical studies performed ex vivo assessments, covering various categories in descending order of emphasis: morphoanatomical and microscopic evaluations, analysis of enzymatic, inflammatory, and biochemical biomarkers, structural parameters, hematological evaluations, and organ index measurements. Several cartilage features were examined in morphoanatomical and microscopic assessments, such as surface integrity, thickness, degeneration, and erosion. Staining techniques, including Hematoxylin/Eosin (H&E), Safranin-O, and Sirius red, were employed for these analyses. Studies have demonstrated that CUR nanoformulations effectively preserve cartilage integrity, reducing inflammatory infiltration, synovial hyperplasia, and bone damage [49].

Regarding inflammatory biomarkers, multiple cytokines were measured, including TNF-α, IL-1β, IL-6, IL-10, and TGF-β, alongside markers such as C-reactive protein, rheumatoid factor, and NF-κB. Following the administration of nanoencapsulated CUR, several inflammatory markers were reduced, while there was an increase in anti-inflammatory cytokines. For instance, a marked decrease was observed in TNF-ɑ, IL-6, IL-1β, and TGF-β levels, coupled with an elevation in IL-10 levels in the nano–CUR treatment group [46].

The assessment of tissue architecture encompassed an analysis of key components such as type II collagen, aggrecan, and a variety of proteoglycans. The nano-based formulations enhanced the expression of structural parameters, offering protection to the cartilage matrix and demonstrating a chondroprotective effect. For example, treating MIA-induced osteoarthritic joints with CUR-loaded micelles [42] yielded a smooth surface with maintained structural integrity and a strong expression of proteoglycan, aggrecan, and collagen. These results indicate the potent anti-inflammatory and anti-arthritic activity of ACP micelles.

In addition, biochemical markers of liver and kidney function, including ALT, AST, ALP, bilirubin, creatinine, uric acid, and urea, were assessed. The lipid profile, comprising total cholesterol, HDL, LDL, and triglycerides, was also measured. Studies have indicated that treatment with CUR in various nano-based formulations improved biochemical parameters related to arthritis. The findings indicated that all groups treated with CUR-loaded nanoparticles exhibited reduced levels of liver and renal function markers in comparison to the gouty group [52]. Concurrently, cholesterol, triglycerides, and LDL levels decreased, while HDL levels rose in all treated groups compared to the MSU group. This indicates hepatic and renal protection and an overall improvement in metabolic status.

Enzymatic and non-enzymatic markers related to the oxidative status and tissue remodeling parameters were also quantified. Nano–CUR treatment decreased oxidative stress and pro-inflammatory enzyme activity, while supporting cartilage protection. For instance, a study demonstrated that the treatment with nanoparticles reduced the malondialdehyde content of the RA group [59]. Another study demonstrated that CUR micelles reduced the expression of matrix metalloproteinases 3 and 9 in articular cartilage, which are involved in tissue remodeling and extracellular matrix degradation, compared to the model group. This finding further confirms the efficacy of CUR nanocarriers in reducing oxidative stress and improving antioxidant status [51]. Kumar and Rai (2012) observed that the migration of leukocytes into the inflamed area was suppressed by proniosomal formulations when compared to standard drug indomethacin, as evidenced by a significant reduction in the total leukocyte count [32]. Lastly, the organ index (thymus, spleen, and liver) was also determined. The work of Zheng and colleagues (2015) evaluated the effect of CUR and methotrexate on the thymus and spleen index [34]. Both treatments resulted in a significant reduction compared to the model group. However, methotrexate had minimal impact on the spleen index, whereas CUR treatment significantly decreased this result, indicating that CUR may promote immune regulation by inhibiting cellular immunity.

### 4.5. Toxicological and Safety Investigations

Nanomaterials can alter substances’ physical, chemical, and toxicological properties, making it impossible to predict their behavior based solely on studies of the original material. Consequently, materials may exhibit toxicological activity, or even enhanced activity, in a manner that is unpredictable when they are introduced to the body in nanoscale forms. This variability can manifest both quantitatively and qualitatively. Therefore, it is necessary to investigate the potential toxicological effects of any material in nano-based formulations before its introduction to biological systems [142]. Only a limited number of the 34 reviewed articles investigated toxicological effects, revealing an important gap in existing scientific literature. Studies encompassing histopathological evaluations, cutaneous assessments, systemic biochemical analyses, vital organ function evaluations, cytotoxicity, and cell viability assessments were undertaken. The findings indicated that most nanocarriers were safe. Nevertheless, one study reported adverse gastrointestinal effects.

To investigate safety, a biocompatibility assessment through histopathological analysis of synovial tissues (H&E) from healthy rat knees following intra-articular injection of CUR-NLC gel was conducted [41]. The findings indicated no signs of macroscopic swelling, joint lesions, or inflammation in the knee. Furthermore, CUR-NLC gel application restored the joint conditions comparable to the control group, highlighting its potential for treating RA. Similarly, Sana and colleagues (2021) showed no significant changes in the epidermis’ chemical composition or irritative effect in the Draize test after CUR-loaded transferosome gel application, suggesting its biocompatibility [23]. Additional research also demonstrated low irritation potential in the selected niosomes and microemulsion formulations through skin irritation tests conducted on rabbits and mice, respectively [32,45]. Okpalaku and coworkers (2023) showed that all formulations maintained an appropriate pH range for skin application, exhibiting slightly acidic to nearly neutral pH values, ensuring no irritation [50]. Two studies have investigated the cytotoxicity and hemocompatibility of various nano-based formulations [21,47], indicating that CUR nanoparticles exhibited biocompatibility and low cytotoxicity and displayed favorable hemocompatibility.

Ultimately, five studies performed a toxicological evaluation of various organs and biochemical parameters. The study conducted by Wang and colleagues (2024) applied H&E staining to assess the main organs, including the heart, liver, spleen, lung, and kidney [51]. Their findings revealed no evidence of damage or significant histological alterations in the micelle-treated groups. Similarly, other work performed histological examinations that also did not show any apparent structural changes in the primary tissues—brain, liver, kidney, intestine, heart, lung, spleen, and muscle [42]. Additionally, they measured serum ALT activity and did not find significant differences compared to the control group, indicating that the prepared micelles have a safety profile. In a related study, some organs (heart, liver, spleen, lung, and kidney) and blood markers were evaluated, and no alterations were observed after the treatment with CUR nanocapsules [54]. Kiyani and colleagues (2019) demonstrated that turmeric nanoparticles possess significant anti-gout therapeutic potential while exhibiting a favorable toxicity profile [39]. This conclusion is supported by comprehensive biochemical analyses and histopathological assessments of tissues (liver, kidneys, and skeletal muscle). However, it is noteworthy that one of the clinical studies described frequent gastrointestinal adverse effects during the treatment with nano-based CUR [53].

### 4.6. Risk of Bias Evaluation

The risk of bias assessment using the SYRCLE tool revealed that most studies presented an uncertain risk in the domains related to selection bias. Although some works indicated the randomization of the allocation sequence, no study specified the method used, thereby hindering a more accurate assessment. Disease induction was carried out consistently across groups, but there was a lack of information on the balance between relevant baseline characteristics. Furthermore, the description of concealed allocation was absent in most studies, except in the study of Sana and collaborators (2021) [23], which presented a low risk of bias. Regarding performance bias, it was observed that most studies neither provided sufficient information about the random allocation of animals nor addressed the blinding of caretakers and researchers. Remarkably, a low risk of bias for this criterion was achieved in just one study [25], indicating that the remaining studies had an uncertain classification in these domains. Regarding the domains related to detection bias, no information was reported on the random selection of animals for outcome assessment and the blinding of evaluators in most studies. The omission of critical information led to an overall uncertain risk of bias across most studies. However, eight studies reported a low risk regarding the blinding of evaluators, suggesting a robust methodological approach in that aspect. In terms of attrition bias, all studies were classified as uncertain, primarily due to the absence of detailed reporting on the inclusion of all animals in analyses and whether losses during the experiments were related to outcomes or distributed evenly across groups. Additionally, there was a notable lack of information concerning the implementation of suitable imputation methods for missing data. Conversely, reporting bias was assessed as low risk across all studies, even in the absence of formal study protocols, as the anticipated outcomes were sufficiently documented. Importantly, all studies were deemed to have a low risk of bias in domain 10, which encompasses other potential sources of bias, indicating that they appeared relatively free from external influences, including those from funders. Therefore, most studies exhibited an unclear risk of bias primarily due to inadequate transparency and missing information.

Comparably, the application of the RoB 2 tool for assessment of risk of bias in clinical trials was used to evaluate the quality of the articles reviewed. In domain 1, which concerns bias arising from the randomization process, all studies exhibited a low risk of bias due to the robust implementation of randomization. However, it is noteworthy that only Lustberg and colleagues (2024) did not describe the method employed for this process [53]. All studies were double-blind, and the allocation concealment was likely maintained until the interventions were assigned. Additionally, the study of Ahmadi and coworkers (2020) explicitly confirmed that there was indeed concealment at that time [43]. Moreover, no imbalances among groups were observed, since the number of participants was similar, indicating adequate randomization. All studies presented a low risk in the second domain, which addresses bias due to deviations from intended interventions. This was because participants, caregivers, and applicators remained unaware of the interventions throughout the studies.

Regarding domain 3, which pertains to bias due to missing outcome data, a low risk was determined, because a substantial amount of data were available [43,53]. On the other hand, Javadi and collaborators (2019) were assigned a high risk of bias, due to the unavailability of most data, not providing sufficient information to assess the possibility of bias resulting from this absence [40]. In domain 4, which focuses on bias in the measurement of the outcome, all studies demonstrated a low risk. The use of validated measurement methods facilitated this outcome, applied consistently across the groups, suggesting that knowledge of the location is unlikely to have influenced the results. Finally, in domain 5, which concerns bias in the selection of the reported result, all studies were evaluated as having a low risk, as they were appropriately registered, outcomes were predefined before data analysis and adhered to a statistical plan. Overall, the results indicated a low risk of bias in two studies [43,53], and a high risk in Javadi and collaborators (2019), due to missing outcome information [40] regarding reasons for participant withdrawal.

The review highlights significant methodological limitations in the included studies evaluated through the SYRCLE and RoB 2 assessment tools. Most preclinical studies exhibited either unclear or high risk of bias in critical areas such as random sequence generation, allocation concealment, and blinding in outcome assessment. In many instances, these concerns stem from inadequate reporting rather than inherent methodological flaws, yet they still compromise the internal validity of the findings. This is particularly concerning subjective outcomes, such as behavioral assessments and histopathological evaluations, which are inherently vulnerable to observer bias. Moreover, the lack of thorough safety evaluations and the limited number of clinical trials further weaken the robustness of the conclusions drawn.

It is relevant to recognize that the methodological rigor of the studies included may impact the synthesis of findings in this review. The prevalence of unclear or high-risk domains, particularly within preclinical research, undermines the reproducibility of results and diminishes the overall strength of the evidence. This issue is especially pertinent for endpoints with inherently low evidence power, such as behavioral or mechanistic measures, which are vulnerable to experimental variability and often lack standardized assessment protocols. Thus, while numerous studies have reported positive pharmacological outcomes associated with CUR-loaded nanocarriers, the interpretation of these results warrants careful consideration. This synthesis represents the current state of the field rather than definitive evidence of efficacy. These observations underscore an urgent need for enhanced methodological standardization, stringent reporting practices, and transparent risk-of-bias mitigation to facilitate more robust and clinically relevant conclusions in future investigations.

### 4.7. Expert Opinion and Perspectives

In our synthesis, we identified the notable prevalence of specific nanocarrier-model combinations, particularly the application of nanoparticles in CFA- or collagen-induced arthritis models. These pairings likely reflect their translational relevance, given that these models effectively replicate key characteristics of human RA, and nanoparticles are favored due to their advantageous physicochemical and biological characteristics. However, despite their frequent use, we cannot conclusively assert their superiority over alternative approaches. This limitation arises primarily from the absence of standardization across studies, the significant or ambiguous risk of bias observed in preclinical evaluations, and the varied outcome measures employed. Such variability constrains the reproducibility and comparability of results. Therefore, based on these scenarios, this scoping review aimed to present a comprehensive overview of the current landscape, mapping the concentrations of evidence and pinpointing research deficiencies. This mapping approach facilitates the identification of future research priorities, which include broadening the scope to encompass alternative models, exploring new formulation types, and adopting standardized and reproducible methodologies to enhance the translational capabilities of the field.

A significant limitation identified in the reviewed literature is the inadequate or absent reporting of correlations between the physicochemical characteristics of nanocarriers and their pharmacological or biological outcomes. Although many studies provided data on key parameters such as particle size, zeta potential, PDI, and encapsulation efficiency, few explored how these attributes impact critical outcomes like drug release kinetics, biodistribution profiles, therapeutic efficacy, or safety. This lack of integrated analysis hinders our ability to derive mechanistic insights into how specific properties of nanocarriers may enhance or detract from their therapeutic performance in various arthritis models. Furthermore, the pronounced methodological heterogeneity across the included studies significantly impedes the feasibility of conducting comparative or correlative analyses. There was considerable variation in the choice of animal models (e.g., rats, mice, rabbits), methods of arthritis induction (such as CFA, collagen, trauma, and urate), routes of administration (oral, intra-articular, topical, intravenous), dosing regimens, and types of biological assessments (histological, behavioral, biochemical, and molecular endpoints). This diversity renders it methodologically unsound to compare outcomes across studies or implement normalization strategies aimed at identifying trends between nanocarrier characteristics and therapeutic outcomes.

Another relevant point is the inadequate assessment of long-term toxicity and safety profiles associated with nanocarrier-based formulations within the selected articles. Given the chronic nature of arthritis and the likely prolonged administration of these therapies, most preclinical studies have primarily concentrated on short-term evaluations, frequently focusing on local irritation or histopathological analysis of select organs. There is a conspicuous absence of systematic investigations into potential long-term effects, including but not limited to genotoxicity, bioaccumulation, and systemic toxicity. Furthermore, the clinical trials reviewed failed to report any toxicological or safety endpoints. This lack of comprehensive toxicological assessment reveals a critical gap in the translational pathway for these nano-based formulations. It emphasizes the urgent need for future research to implement standardized long-term toxicity protocols as integral components of safety evaluations.

Considering the findings from the articles, it is essential to underscore the gap in clinical studies identified throughout the review process. Specifically, only 3 out of the 34 articles reviewed were classified as clinical trials. This limitation highlights the need for greater emphasis on clinical research to enhance our understanding of human arthritis. Thus, while preclinical models play a critical role in comprehending new therapeutic strategies, they do not fully replicate the complexity inherent in human physiopathology. Additionally, many of the preclinical studies reviewed lacked essential information, including the concentration of CUR, encapsulation efficiency, in vitro drug release studies, and safety assessments. These limitations hindered the comparative analysis of results, requiring careful consideration, particularly concerning the translation of these findings into clinical practice. In addition, it was noted that only a limited number of behavioral assessments were conducted, as less than half of the reviewed studies addressed this aspect. Furthermore, as previously addressed, there is a considerable need for greater detail and clarity in the descriptions of treatment protocols, since many studies do not specify the duration or frequency, which creates challenges in data comparison, as demonstrated by the SYRCLE tool. Thus, as most preclinical studies showed an uncertain risk of bias, their conclusions may have been compromised.

Based on the synthesized evidence, several key research gaps have been identified in the development of therapeutic strategies for joint diseases. Notably, there is a significant paucity of studies assessing solid or semisolid formulations (hydrogels and films) despite their potential benefits for topical or intra-articular delivery. The current clinical evidence remains limited, as evidenced by the inclusion of only three randomized controlled trials, underscoring the pressing need for translational research and methodologically robust clinical trials. Moreover, advanced drug delivery systems, including exosomes, nanosponges, and hybrid lipid–polymer nanoparticles, have yet to be thoroughly investigated within arthritis models, despite their favorable physicochemical and biological characteristics. Additionally, safety assessments across the existing literature have been inadequately reported, hindering the translational applicability of these formulations. Future research should prioritize addressing these gaps through rigorous experimental designs, including direct comparisons between free and nano-based CUR formulations. Emphasis should also be placed on pharmacokinetics, tissue distribution, and standardized outcome measures. These investigations will be critical for the rational development of optimized CUR-based nanotherapeutics for the treatment of arthritis.

## 5. Conclusions

This scoping review synthesizes essential information from articles that evaluated the use of nanotechnology in the CUR administration for arthritis treatment. The reviewed literature highlights the potential of CUR encapsulated within nanocarrier systems for the treatment of arthritis, enhancing the pharmacological effectiveness of this bioactive compound. Overall, most of the studies showed enhanced performance of nano-based formulations compared to free CUR and potentially demonstrated effects that are similar to or even better than those of traditional anti-inflammatory medicines. However, to facilitate understanding and comparison of research findings, greater clarity and comprehensive information during the elaboration of studies are essential. Standardization of experimental protocols is crucial to ensure consistent inclusion of detailed methodologies for the induction and treatment of animals. Moreover, a comparison between the effects of free CUR and its nanocarriers, along with thorough toxicological and safety assessments, is essential to enhance the quality of the literature, particularly when clinical investigations are conducted.

Future research should focus on comprehensive comparative analyses to address the existing knowledge gap regarding nano-based formulations and their benefits. Additionally, there is a pressing need for in-depth toxicological assessments and safety evaluations, coupled with extensive clinical trials, to substantiate the efficacy of this nanotechnology. Although current research findings are promising, a rigorous and standardized approach to investigation is vital to validate the translational potential of CUR-loaded nanocarriers as an effective therapeutic strategy for arthritis.

## Figures and Tables

**Figure 2 pharmaceutics-17-01022-f002:**
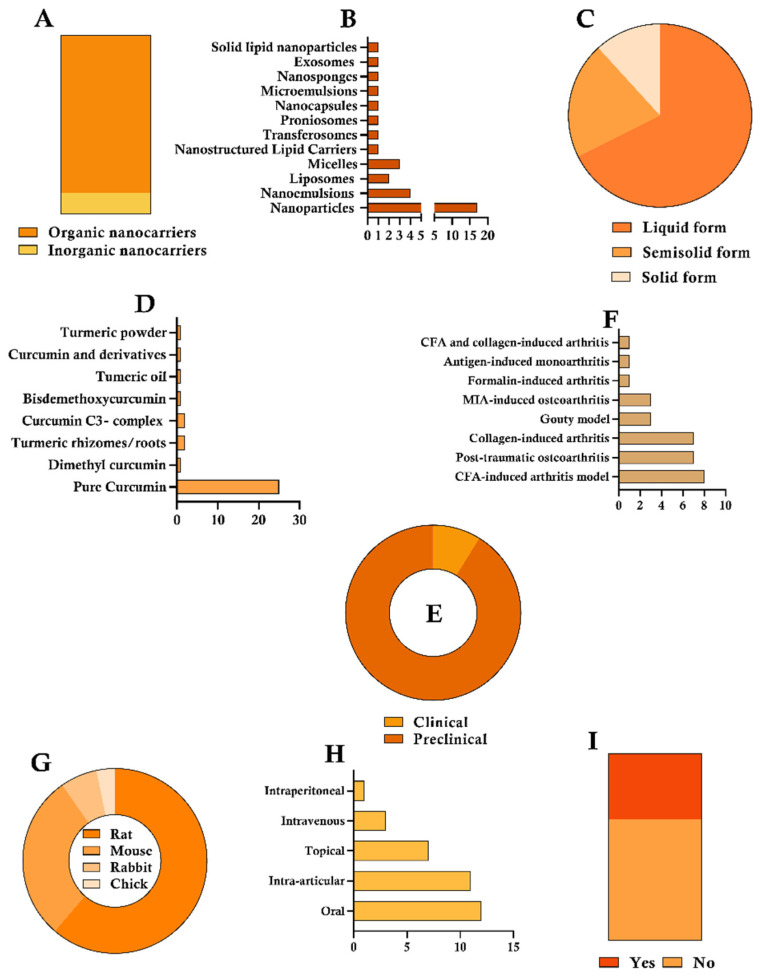
Descriptive statistical analysis was performed in the present study, focusing on the following parameters: (**A**) Classification of nanocarriers as organic or inorganic, (**B**) Specific type of nanocarrier employed, (**C**) Composition of the nanocarrier–curcumin system, (**D**) Formulation or form of curcumin utilized, (**E**) Study design, (**F**) Type of inducer applied, (**G**) Animal model adopted in preclinical studies, (**H**) Route of administration of the nanocarrier–curcumin complex, and (**I**) Whether safety assessment was conducted.

**Figure 3 pharmaceutics-17-01022-f003:**
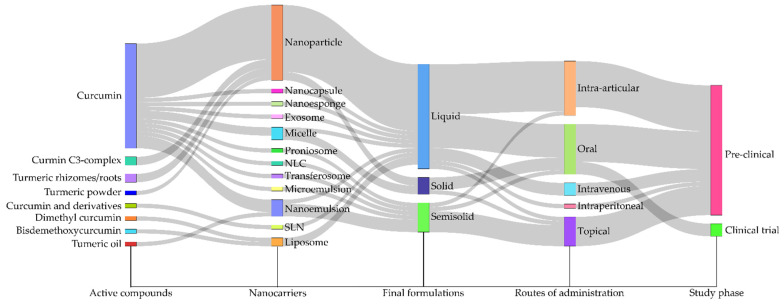
Sankey diagram of the available evidence on CUR-containing nano-based systems for arthritis management. The analysis includes different types of CUR sources, types of nanocarriers, pharmaceutical forms, routes of administration, and study designs (preclinical studies and clinical trials). Abbreviations: Solid lipid nanoparticles (SLNs); Nanostructured lipid carriers (NLCs).

**Figure 4 pharmaceutics-17-01022-f004:**
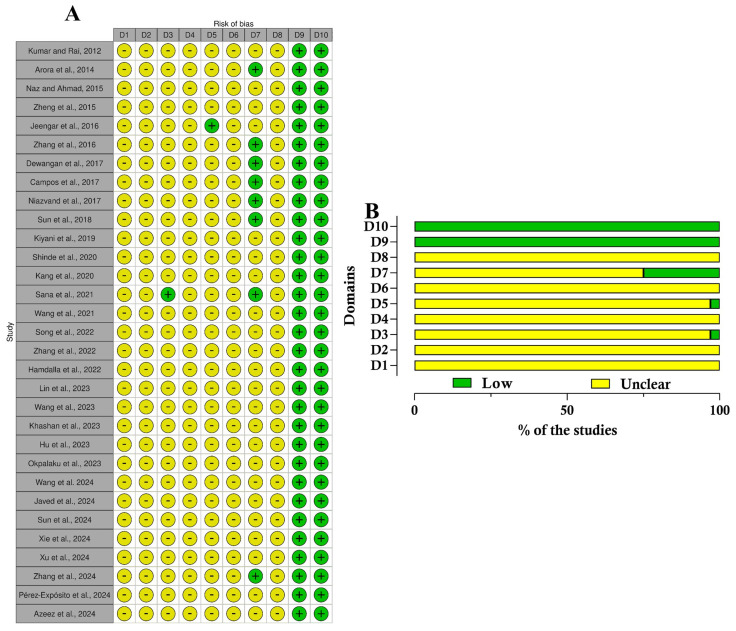
Risk of bias summary using the SYRCLE tool. (**A**) Scores of each paper per domain (D): 1—Sequence generation; 2—Baseline characteristics; 3—Allocation concealment; 4—Random housing; 5—Blinding; 6—Random outcome assessment; 7—Blinding; 8—Incomplete outcome data; 9—Selective outcome reporting; 10—Other sources of bias. (**B**) Overall scores.

**Figure 5 pharmaceutics-17-01022-f005:**
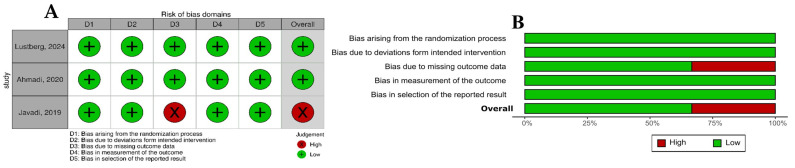
Risk of bias summary using RoB 2: (**A**) Scores of each paper per domain; (**B**) Overall scores.

**Figure 6 pharmaceutics-17-01022-f006:**
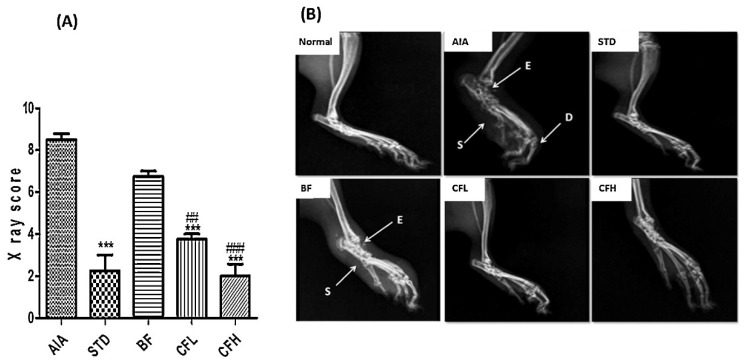
(**A**) Radiological assessment of adjuvant arthritic and treated rats. All values are expressed as mean ± SEM (*n* = 6). *** *p* < 0.001 compared to the AIA control; ### *p* < 0.001, ## *p* < 0.01 compared to the standard group; (**B**) Radiographic findings in ankle joints. Key: S = soft tissue swelling; D = deformity; E = erosion. Abbreviations: AIA—adjuvant-induced arthritis; BF—Blank gel formulation; CFH—CUR nanoemulsion with 0.5% *w*/*w* curcumin; CFL—CUR nanoemulsion with 0.25% *w*/*w* CUR; STD—Standard group treated with indomethacin. The rights to utilize this image were obtained directly from the journal. The source of this article is accessible via DOI: 10.1016/j.ijpharm.2016.04.052 [25].

**Figure 7 pharmaceutics-17-01022-f007:**
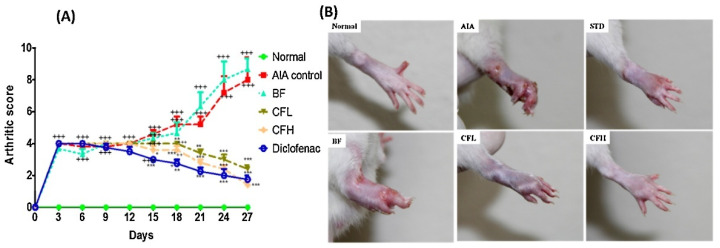
(**A**) Evaluation of disease progression in arthritic animals following treatment with an emu oil-based curcumin nanoemulsion gel was conducted using clinical scoring for arthritis. All values are presented as mean ± SEM (*n* = 6). +++ *p* < 0.001 vs. normal group; *** *p* < 0.001, ** < 0.01, * < 0.5 vs. AIA control group. The severity of arthritis was assessed based on the degree of erythema and swelling observed in all four paws, using the scoring system. (**B**) Representative photographs of the lateral view of the right hind paw from animals treated with the formulation, captured on the 28th day following FCA injection. Abbreviations: AIA—adjuvant-induced arthritis; BF—Blank gel formulation; CFH—curcumin nanoemulsion with 0.5% *w*/*w* curcumin; CFL—curcumin nanoemulsion with 0.25% *w*/*w* CUR. The rights to utilize this image were obtained directly from the journal. The source of this article is accessible via DOI: 10.1016/j.ijpharm.2016.04.052 [25].

**Figure 8 pharmaceutics-17-01022-f008:**
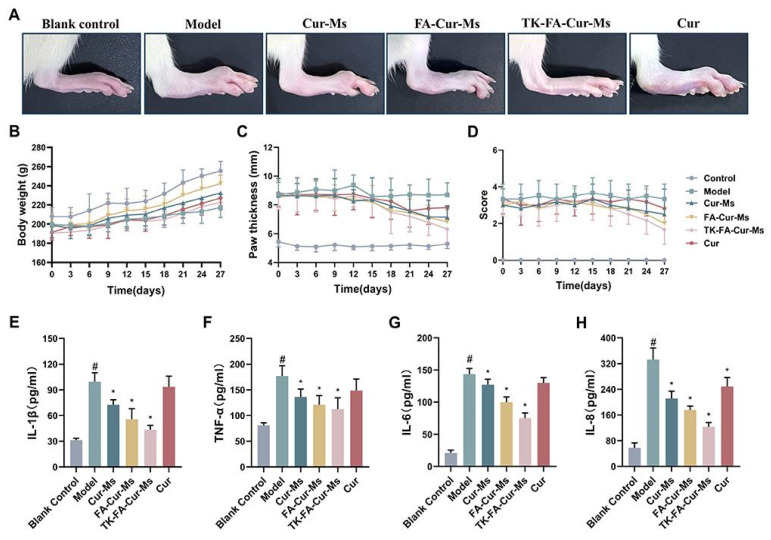
Anti-arthritic efficacy in CIA rats. (**A**) Representative photographs of hind paws from each group. (**B**) Body weight of arthritic rats over time of varying formulations. (**C**) Paw thickness of arthritic rats over time of varying formulations. (**D**) Arthritis index score of arthritic rats over time of varying formulations. (**E**–**H**) The levels of IL-1β, TNF-α, IL-6, and IL-8 in the serum of CIA rats. #, vs. Blank Control; *, vs. Model. *p* < 0.05. Data are presented as the mean SD (*n* = 6). Abbreviations: CIA—Collagen-induced arthritis; TK-FA-Cur-Ms—ROS-responsive folate-modified CUR micelles; FA-Cur-Ms—folate-modified curcumin micelles. The rights to utilize this image were obtained directly from the journal. The source of this article is accessible via DOI: 10.2147/IJN.S458957 [51].

**Table 1 pharmaceutics-17-01022-t001:** General physicochemical characteristics of the selected studies.

Author	Nanocarrier	Composition	Preparation Method	Final Form
Kumar and Rai, 2012 [32]	Proniosomes	Organic	Ether injection method	Semisolid
Arora et al., 2014 [24]	Solid lipid nanoparticle	Organic	Hot homogenization and a melt ultrasonication	Liquid
Naz and Ahmad, 2015 [33]	Nanoemulsion	Organic	Spontaneous emulsification	Semisolid
Zheng et al., 2015 [34]	Nanoemulsion	Organic	High-pressure homogenization method	Semisolid
Jeengar et al., 2016 [25]	Nanoemulsion	Organic	Spontaneous emulsification	Semisolid
Zhang et al., 2016 [35]	Nanoparticle	Organic	Ultrasonification	Solid
Dewangan et al., 2017 [22]	Nanoparticle	Organic	Nanoprecipitation	Liquid
Campos et al., 2017 [36]	Nanoparticle	Inorganic	Conjugation	Liquid
Niazvand et al., 2017 [37]	Nanoparticle	Organic	Solvent solid-in-oil-in-water emulsion evaporation technique	Liquid
Sun et al., 2018 [38]	Liposome	Organic	Thin-film method	Liquid
Kiyani et al., 2019 [39]	Nanoparticle	Organic	Ultrasonification	Liquid
Javadi et al., 2019 [40]	Micelle	Organic	Patent	Solid
Shinde et al., 2020 [41]	Nanostructured lipid carrier	Organic	Hot homogenization and a subsequent melt ultrasonication	Semisolid
Kang et al., 2020 [42]	Nanoparticle	Organic	Self-emulsification	Liquid
Ahmadi et al., 2020 [43]	Micelle	Organic	Patent	Solid
Sana et al., 2021 [23]	Transferosome	Organic	Thin-film hydration	Semisolid
Wang et al., 2021 [20]	Liposome	Organic	Thin-film method	Liquid
Song et al., 2022 [44]	Nanoparticle	Organic	Sonication and reversible noncovalent interactions	Liquid
Zhang et al., 2022 [45]	Microemulsion	Organic	Spontaneous emulsification	Semisolid
Hamdalla et al., 2022 [46]	Nanoparticle	Organic	Nanoprecipitation	Liquid
Lin et al., 2023 [47]	Nanoparticle	Organic	Nanoprecipitation	Liquid
Wang et al., 2023 [21]	Nanoparticle	Organic	Nanoprecipitation	Liquid
Khashan et al., 2023 [48]	Nanoparticle	Inorganic	Green chemistry	Liquid
Hu et al., 2023 [49]	Nanoparticle	Organic	Sonication and reversible noncovalent interactions	Liquid
Okpalaku et al., 2023 [50]	Nanoemulsion	Organic	Spontaneous emulsification	Liquid
Wang et al., 2024 [51]	Micelle	Organic	Thin-film dispersion	Liquid
Javed et al., 2024 [52]	Nanoparticle	Organic	Nanoprecipitation	Liquid
Lustberg et al., 2024 [53]	Nanoparticle	Organic	Patent	Solid
Sun et al., 2024 [54]	Nanocapsules	Organic	Inborn microcrystallization method	Liquid
Xie et al., 2024 [55]	Nanoparticle	Inorganic	Ultrasound assisted synthesis	Liquid
Xu et al., 2024 [56]	Nanosponge	Organic	Nanoprecipitation	Liquid
Zhang et al., 2024 [57]	Exosome	Organic	Ultrasound encapsulation	Liquid
Pérez-Expósito et al., 2024 [58]	Nanoparticle	Organic	-	Liquid
Azeez et al., 2024 [59]	Nanoparticle	Inorganic	Co-precipitation	Liquid

**Table 2 pharmaceutics-17-01022-t002:** Experimental design of the selected studies.

Author	CUR Form and Administration Route	Study Design	Safety Assessment
Kumar and Rai, 2012 [32]	Pure drug, topical administration	CFA-induced arthritis rat model	Skin irritancy test on male albino rabbits
Arora et al., 2014 [24]	95% cur and 5% methoxycurcumin + bis-methoxycurcumin, oral administration	CFA-induced arthritis rat model	-
Naz and Ahmad, 2015 [33]	Pure drug, intra-articular administration	CFA-induced arthritis rat model	-
Zheng et al., 2015 [34]	Pure drug, oral administration	CFA-induced arthritis rat model	-
Jeengar et al., 2016 [25]	Pure drug, topical administration	CFA-induced arthritis rat model	-
Zhang et al., 2016 [35]	Pure drug, topical administration	Post-traumatic osteoarthritis mouse model	-
Dewangan et al., 2017 [22]	Pure drug, oral administration	CFA-induced arthritis rat model	-
Campos et al., 2017 [36]	Pure drug, intra-articular administration	Post-traumatic osteoarthritis mouse model	-
Niazvand et al., 2017 [37]	Pure drug, oral administration	Monoiodoacetate-induced osteoarthritis in rats	-
Sun et al., 2018 [38]	Dimethyl curcumin (DiMC), intra-articular administration	Collagen-induced arthritis rat model	-
Kiyani et al., 2019 [39]	Turmeric powder, oral administration	Monosodium urate-induced Gouty mouse model	Clinical parameters, biochemical analyses, and histopathological evaluation
Javadi et al., 2019 [40]	C3-complex form of curcumin, oral administration	Randomized, double-blind, controlled trial with RA patients	-
Shinde et al., 2020 [41]	Pure drug, intra-articular administration	Antigen-induced monoarthritis model in rats	Knee histopathological studies
Kang et al., 2020 [42]	Pure drug, intra-articular administration	Monoidoacetic acid (MIA)-induced knee osteoarthritis—mouse	Hepatic markers
Ahmadi et al., 2020 [43]	C3-complex form of curcumin, oral administration	Randomized, double-blind, placebo-controlled clinical trial with patients clinically diagnosed with AS	-
Sana et al., 2021 [23]	Pure drug, topical administration	CFA-induced arthritis mice model	In vivo Draize skin irritation and histopathological studies
Wang et al., 2021 [20]	BDMC, oral administration	Potassium oxonate induced Gouty rat model	-
Song et al., 2022 [44]	Pure drug, intraperitoneal administration	Collagen-induced arthritis rat model	-
Zhang et al., 2022 [45]	Pure drug, topical administration	Collagen-induced arthritis mouse model	Skin irritation test in mice
Hamdalla et al., 2022 [46]	Pure drug, intra-articular administration	Monoidoacetic acid (MIA)-induced knee osteoarthritis	-
Lin et al., 2023 [47]	Pure drug, intra-articular administration	Post-traumatic osteoarthritis mouse model	Hemolysis assay
Wang et al., 2023 [21]	Pure drug, intra-articular administration	Post-traumatic osteoarthritis mouse model	-
Khashan et al., 2023 [48]	Turmeric rhizomes in dry form, oral administration	CFA-induced arthritis rat model	-
Hu et al., 2023 [49]	Pure drug, intra-articular administration	Collagen-induced arthritis rat model	-
Okpalaku et al., 2023 [50]	Turmeric oil, topical administration	Formalin-induced arthritis rat model	Skin irritation test in mice
Wang et al., 2024 [51]	Pure drug, intravenous administration	Collagen-induced arthritis (CIA) rat model	Histopathological evaluation
Javed et al., 2024 [52]	Pure drug, oral administration	Monosodium urate-induced Gouty mouse model	-
Lustberg et al., 2024 [53]	Curcumin C3-complex, oral administration	Randomized placebo-controlled, double-blind clinical trial—Aromatase inhibitor-induced arthralgia	Gastrointestinal adverse effects were commonly reported
Sun et al., 2024 [54]	Pure drug, intravenous administration	Collagen-induced arthritis mouse model	Histopathological evaluation
Xie et al., 2024 [55]	Fresh *Curcuma longa* roots, intra-articular administration	Post-traumatic osteoarthritis Chick model	-
Xu et al., 2024 [56]	Pure drug, intra-articular administration	Post-traumatic osteoarthritis rat model	-
Zhang et al., 2024 [57]	Pure curcumin, intravenous administration	Collagen-induced arthritis mouse model	-
Pérez-Expósito et al., 2024 [58]	Pure drug, intra-articular administration	Post-traumatic osteoarthritis rabbit model	-
Azeez et al., 2024 [59]	Pure drug, oral administration	CompleteFreund’s adjuvant and collagen-induced arthritis rabbit model	-

## Data Availability

Not applicable.

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
