# Peer review of "Nanocarriers Containing Curcumin and Derivatives for Arthritis Treatment: Mapping the Evidence in a Scoping Review"

_pharmaceutics, 2025, doi:10.3390/pharmaceutics17081022_

Round 1
Reviewer 1 Report
Comments and Suggestions for Authors
The manuscript "Nanocarriers Containing Curcumin and Derivatives for Arthritis Treatment: Mapping the Evidence in a Scoping Review" is a well prepared review which followed the PRISMA guidelines. It is well justified, well organized and properly presented. The manuscript brings new knowledge and identifies important gaps that need investigation. There are some minor issues that I noticed and below is a list of my comments:
- In my opinion, loading capacity is more significant for comparing the nanoparticles instead of encapsulation efficiency as the second is affected by the initial loading amount of curcumin and the loading capacity could directly compare the nanocarriers.
- The statement that 97% is "remarkable value" (line 363) or "exceptional EE" (line 591) is a little bit exaggerated. In how many of the identified studies the EE is above 95% and close to 97% (at least one more ref. 20 as stated on line 446 has quite similar EE). This is the reason I suggest loading capacity for comparison rather than encapsulation efficiency.
- The mentioned value for highest zeta potential +205 mV (line 361 and line 590) at fist seemed very odd as values above 60-70 are typically not measured. Indeed, it is not properly reported. In the original source (ref. 21) a value of +20.5mV could be found. Please, verify that the highest reported value of zeta potential is indeed in this reference.
- On line 444 the authors mention "positive PDI (0.35+/-0.016). What should "positive" reflect to in this phrase?
- On line 536 an unfinished sentence can be seen: " Complete release of CUR."
- In my opinion, some modification of the paragraph discussing the release behavior is advisable (the text between lines 568 and 580). The observed differences (especially in the case of free CUR) are highly dependent on the release procedure (with or without dialysis membrane, with or without surfactant, e.g. SDS in ref. 33, etc.). This issue should be also clearly stated in the manuscript and properly addressed as a limitation in the comparison between studies. The lack of unified conditions for testing are general problem in all nanocarrier formulations and not only for in vitro release studies.
- Please, verify the meaning of the sentence on line 624: "Lastly, fresh Curcuma longa roots were used to extract Curcuma longa".
- In the light of toxicological studies, were there long-term effects investigated in the identified references. Especially in the case of chronic treatment of arthritis, some types of nanoparticles could have genotoxic or other effects of the type.
Author Response
# REVIEWER 1
The manuscript "Nanocarriers Containing Curcumin and Derivatives for Arthritis Treatment: Mapping the Evidence in a Scoping Review" is a well prepared review which followed the PRISMA guidelines. It is well justified, well organized and properly presented. The manuscript brings new knowledge and identifies important gaps that need investigation. There are some minor issues that I noticed and below is a list of my comments:
Answer: We appreciate the reviewer for the favorable assessment of our manuscript and for acknowledging the significance, structure, and methodological rigor of our scoping review. We are also grateful for the constructive feedback provided. Your observations contributed significantly to improving the scientific clarity, precision, and robustness of the final version of our work. All suggestions were carefully considered and addressed either directly in the revised manuscript or in the specific responses below. The changes made to the text are clearly highlighted in green in the revised version and are annotated with indications of the respective page and line numbers for transparency and ease of verification.
Once again, we thank you for your valuable contribution to the refinement of our study.
In my opinion, loading capacity is more significant for comparing the nanoparticles instead of encapsulation efficiency as the second is affected by the initial loading amount of curcumin and the loading capacity could directly compare the nanocarriers.
Answer: We appreciate the reviewer’s insightful and technically informed observation. We concur that loading capacity (LC%) is a more appropriate metric than encapsulation efficiency (EE%) for comparing and evaluating the performance of different nanocarrier systems. Encapsulation efficiency quantifies the fraction of the initial drug mass that is successfully integrated into the nanocarrier. This metric is highly dependent on the initial drug concentration during formulation; thus, high EE% values can be misleading, as they can be achieved even with a minimal actual drug load if the starting concentration is low. Conversely, loading capacity measures the drug amount relative to the total weight of the final nanocarrier product (drug + carrier), providing a more reliable and standardized parameter for cross-formulation comparisons.
However, our scoping review aimed to map and delineate the landscape of existing evidence rather than conduct a quantitative comparison of nanocarrier systems. Unfortunately, only a limited number of studies included LC% data, and those that did often lacked a consistent definition or employed non-standard calculation methods. In contrast, while EE% was reported more frequently, it also exhibited considerable methodological inconsistency. The variability in analytical protocols, formulation techniques, and reporting practices across studies, including differences in initial CUR amounts, nanocarrier compositions, and calculation methodologies, prevented a systematic extraction or comparison of LC values.
We acknowledge the importance of this issue and have addressed this limitation in the discussion section of our review. There, we highlight the challenges posed by inconsistent reporting and methodological diversity, which impede direct comparisons and underscore the need for standardization in future research efforts. For your review, please check the modifications marked in green in the revised manuscript, in the topic “Expert Opinion and Perspectives” (Manuscript’s page 31).
The statement that 97% is "remarkable value" (line 363) or "exceptional EE" (line 591) is a little bit exaggerated. In how many of the identified studies the EE is above 95% and close to 97% (at least one more ref. 20 as stated on line 446 has quite similar EE). This is the reason I suggest loading capacity for comparison rather than encapsulation efficiency.
Answer: We appreciate the reviewer’s comments on our use of descriptors like "remarkable" or "exceptional". We acknowledge that such terminology may be excessive. As highlighted by the reviewer, EE% by itself does not provide a comprehensive assessment of nanocarrier performance, particularly without accompanying LC% data. It is essential to interpret EE% alongside other critical parameters, including total drug load, formulation stability, and biological efficacy.
Considering this feedback, we have revised the manuscript to either eliminate or temper these descriptors, favoring more precise and impartial language when discussing EE% values. Our aim is to present an accurate, evidence-based discussion that refrains from suggesting the superiority of isolated metrics. We value the reviewer’s vigilance in maintaining scientific rigor.
The mentioned value for highest zeta potential +205 mV (line 361 and line 590) at fist seemed very odd as values above 60-70 are typically not measured. Indeed, it is not properly reported. In the original source (ref. 21) a value of +20.5mV could be found. Please, verify that the highest reported value of zeta potential is indeed in this reference.
Answer: We appreciate the reviewer for pointing out this critical oversight. The zeta potential value cited in reference [21] was indeed misstated; the accurate value is +20.5 mV, not +205 mV. This has been amended in all pertinent sections of the manuscript. Furthermore, we undertook a comprehensive review of all zeta potential values throughout the manuscript to ensure alignment with the original data and to mitigate the risk of further inaccuracies. We are thankful for the reviewer’s meticulous examination, which has significantly enhanced the precision and reliability of our work.
On line 444 the authors mention "positive PDI (0.35+/-0.016). What should "positive" reflect to in this phrase?
Answer: We thank the reviewer for spotting this inconsistency. The term “positive” in this context was a typographical error and does not apply to the polydispersity index (PDI), which is a non-negative numerical value and should not be described in terms of polarity. We have corrected this phrasing in the manuscript and now simply report the PDI value as “0.35 ± 0.016.” (Manuscript’s page 15, Line 449). As with the previous comment, we appreciate the reviewer’s attentive reading and constructive input, which contributed to improving the clarity and technical accuracy of the manuscript.
On line 536 an unfinished sentence can be seen: " Complete release of CUR."
Answer: We thank the reviewer for pointing out this textual issue. The sentence in question was inadvertently left incomplete during the drafting of the discussion section. Upon review, we concluded that the fragment did not add essential information and have therefore removed it to improve the clarity and flow of the paragraph.
We sincerely apologize for this oversight and have conducted a careful re-reading of the full manuscript to identify and correct similar textual inconsistencies. We appreciate the reviewer’s detailed and helpful feedback, which allowed us to polish the final version of the text.
In my opinion, some modification of the paragraph discussing the release behavior is advisable (the text between lines 568 and 580). The observed differences (especially in the case of free CUR) are highly dependent on the release procedure (with or without dialysis membrane, with or without surfactant, e.g. SDS in ref. 33, etc.). This issue should be also clearly stated in the manuscript and properly addressed as a limitation in the comparison between studies. The lack of unified conditions for testing are general problem in all nanocarrier formulations and not only for in vitro release studies.
Answer: We thank the reviewer for this important and well-founded suggestion. We fully agree that the interpretation of in vitro release data is highly dependent on methodological factors, such as the use (or not) of dialysis membranes, the presence of solubilizing agents, agitation speed, release medium composition, and duration of the experiment. These variations significantly affect the comparability of release profiles between studies, especially when comparing free CUR and nanocarrier-encapsulated CUR.
In response to this valuable comment, we have revised the indicated paragraph to acknowledge this limitation clearly and to emphasize the need for standardized protocols in evaluating nanocarrier release kinetics. Furthermore, we extended the discussion to recognize that the lack of methodological harmonization is a broader issue affecting not only release studies but also other stages of nanocarrier development, including characterization, biological evaluation, and safety assessment.
We thank the reviewer again for this thoughtful contribution, which helped us reinforce a critical methodological gap in the field. The modifications performed have been marked in green in the revised manuscript (Manuscript’s page 17, Line 576 to 600).
Please, verify the meaning of the sentence on line 624: "Lastly, fresh Curcuma longa roots were used to extract Curcuma longa".
Answer: We appreciate the reviewer for bringing this oversight to our attention. The sentence in question was mistakenly worded during the editing phase. The intended message was to specify that fresh Curcuma longa rhizomes were utilized to prepare a crude extract. We have revised the statement for precision and clarity (Manuscript’s page 18, Line 638 to 639).
In the light of toxicological studies, were there long-term effects investigated in the identified references. Especially in the case of chronic treatment of arthritis, some types of nanoparticles could have genotoxic or other effects of the type.
Answer: We appreciate the reviewer highlighting this critical observation, particularly given the chronic nature of arthritis and the associated risks of long-term nanocarrier system usage. This feedback underscores a significant shortfall in the existing preclinical and clinical research that we acknowledge needs to be addressed.
Among the 34 studies analyzed in this review, only 11 included some form of toxicological evaluation. These assessments tended to be narrow in focus, primarily addressing short-term endpoints such as:
- Skin irritation tests are conducted in rabbits or mice.
- Hemolysis assays to assess blood compatibility.
- Hepatic biochemical markers (e.g., ALT, AST) indicating liver function.
- Histopathological evaluations of the knee joint and skin.
- General clinical observations and behavioral assessments.
- Occasional reporting of gastrointestinal adverse effects.
While these evaluations are valuable, they are insufficient for identifying long-term, cumulative, or genotoxic impacts, which are essential considerations for chronic administration scenarios. Moreover, the clinical studies included in this review did not report any toxicological or safety-related endpoints, further constraining our comprehension of long-term safety in human subjects. Recognizing this significant evidence gap, we have explicitly addressed it in the discussion section of the manuscript. We concur with the reviewer that future research endeavors must integrate robust and standardized protocols for chronic toxicity evaluation, including assessments of genotoxicity, bioaccumulation, and immunotoxicity, particularly for nanocarriers designed for prolonged use in inflammatory or degenerative conditions.
The modifications performed are marked in green in the revised manuscript. (Manuscript’s page 31, Line 1282 to 1293).

Reviewer 2 Report
Comments and Suggestions for Authors
Although the review presents a certain scientific interest, there are some important comments:
- Describe how the quality of the studies affected synthesis—the current mention of SYRCLE and RoB 2 appraisals is indicative but sub-integrated into the discussion or findings.
- Describe why some nanoformulation-model pairings are more frequent.
- The review appropriately distinguishes between CUR, BDMC, and turmeric preparations but often pools results from these sources without acknowledging their pharmacologic non-equivalence.
- Risk of bias appraisals using SYRCLE and RoB 2 tools were not applied strictly. Most studies were marked as "unclear" or "high risk," but authors ignored these outcomes in interpreting the data.
- Studies covered are very heterogeneous in terms of curcumin forms, type of nanocarriers, routes of administration, and model animals. However, heterogeneity is not supported by the review without any subgroup analysis, normalization of results, or weightage based on relevance.
- The review is mostly descriptive with a focus on reporting formulation characteristics such as zeta potential and particle size and less on identifying if these correlate with efficacy, safety, or biological activities.
- The review doesn't state any gap-driven research or hypothesis questions. It doesn't provide suggestions for future directions of research or specific nanocarrier systems to be studied in the future.
- The English language requires further improvement.
The English language requires further improvement.
Author Response
# REVIEWER 2
Although the review presents a certain scientific interest, there are some important comments:
Answer: We extend our sincere gratitude to the Reviewer for the meticulous review and insightful critique of our manuscript. Your observations and recommendations were instrumental in enhancing the analytical rigor, conceptual clarity, and overall depth of our scoping review.
We addressed all the points raised, implementing specific revisions in the manuscript and providing comprehensive justifications in our response letter. Notably, we elaborated on methodological limitations, the heterogeneity within the studies, translational relevance, and the pharmacological distinctions among curcumin derivatives, as well as the current gaps in toxicological assessments, which are critical aspects you appropriately underscored.
All modifications made to the manuscript are distinctly highlighted in green in the revised version, with precise references to page and line numbers for ease of verification.
We truly appreciate your insightful contributions, which have significantly elevated the scientific quality of our work.
- Describe how the quality of the studies affected synthesis—the current mention of SYRCLE and RoB 2 appraisals is indicative but sub-integrated into the discussion or findings.
Answer: We appreciate the insightful suggestion from the reviewer. As noted, our initial presentation of risk of bias assessments using the SYRCLE and RoB 2 tools was descriptive, and the implications for evidence synthesis and interpretation were not thoroughly integrated into the discussion.
To rectify this, we have expanded our discussion to elucidate how the identified methodological limitations influenced both the synthesis process and the interpretation of our findings. The significant proportion of studies demonstrating unclear or high risk of bias, particularly concerning randomization, blinding, and allocation concealment, impairs the reproducibility of results and diminishes the overall strength of the evidence. These limitations are particularly pronounced in mechanistic and behavioral outcomes within preclinical studies, which are inherently more vulnerable to subjective biases and experimental variability. Thus, such limitations undermine the robustness of reported therapeutic effects and complicate the identification of consistent patterns across studies.
Furthermore, given the predominance of exploratory preclinical research, we acknowledge that our review reflects the current state of knowledge rather than high-certainty evidence. We emphasize the urgent need for advancement toward more standardized experimental protocols, enhanced transparency in reporting, and improved methodological rigor. Such improvements are crucial not only to facilitating reproducibility and future meta-analytic aggregations but also for bolstering translational research efforts.
For your review, please check the modifications marked in green in the revised manuscript, in the topic “Expert Opinion and Perspectives” (Manuscript’s page 31).
- Describe why some nanoformulation-model pairings are more frequent.
Answer: We thank you for your observation. It is recognized that certain combinations of nanoformulations and experimental models, specifically, those involving nanoparticles in CFA- or collagen-induced arthritis models, are disproportionately represented in the reviewed literature. Considering the reviewer’s feedback, we have broadened the discussion to delve into potential reasons for these recurring pairings while acknowledging the constraints that impede definitive conclusions regarding their relative superiority or preference.
Nanoparticles constitute the predominant nanocarrier system, appearing in 17 of the 34 studies analyzed, while CFA- and collagen-induced arthritis models were utilized in 8 and 7 studies, respectively. This trend can be attributed in part to the translational relevance of these models, which replicate key characteristics of rheumatoid arthritis, including synovial inflammation, joint swelling, and systemic immune response. Furthermore, nanoparticles, particularly those derived from biodegradable polymers like PLGA and chitosan, are well-established in this research domain due to their versatility, biocompatibility, and capacity to enhance the solubility and stability of bioactive compounds like curcumin. Collectively, these elements provide a pragmatic framework for proof-of-concept studies focused on anti-inflammatory and anti-arthritic interventions.
Nonetheless, it is crucial to highlight that the frequent application of these models and formulations is tempered by methodological heterogeneity, and an overall high or indeterminate risk of bias as indicated by SYRCLE assessments. The absence of standardized outcome measures, coupled with variability in dosing regimens, administration routes, and incomplete documentation of physicochemical characteristics, complicates efforts to ascertain whether these combinations are genuinely the most efficacious or simply the most conventional and accessible.
Consequently, this scoping review does not seek to rank nanoformulation-model pairings based on their performance metrics. Rather, its primary objective is to outline the prevailing research landscape, pinpoint areas of research concentration, and identify evidence gaps associated with under-investigated combinations, administration routes, or formulation types (e.g., topical semisolids, exosomes). These observations underscore the pressing need for enhanced methodological rigor, standardization, and diversification in future studies to facilitate stronger comparative and translational insights.
These modifications have been marked in green in the revised manuscript. (Manuscript’s page 31, Line 1249 to 1264).
- The review appropriately distinguishes between CUR, BDMC, and turmeric preparations but often pools results from these sources without acknowledging their pharmacologic non-equivalence.
Answer: We appreciate the reviewer’s insightful observation. The studies included in our analysis utilized a range of curcumin sources, such as pure curcumin (CUR), bisdemethoxycurcumin (BDMC), C3-complex, turmeric oil, and whole rhizome extracts. Although we differentiated these sources during data extraction and in our descriptive synthesis, we recognize that the pharmacological disparities among these curcuminoids and turmeric derivatives were not sufficiently highlighted in certain sections of our discussion.
To enhance clarity, we have revised the discussion to include a detailed acknowledgment of the pharmacologic non-equivalence between these compounds. While they possess structural similarities and exhibit overlapping mechanisms of action, their differences in potency, bioavailability, metabolic stability, and molecular targets are significant. Additionally, the variability in purity, source, and analytical methods across the studies complicates the interpretation of pooled outcomes. We have emphasized that, given these discrepancies and the inherent lack of standardization in the reviewed studies, any generalizations regarding their efficacy should be approached with caution.
These modifications have been marked in green in the revised manuscript. (Manuscript’s page 19, Line 683 to 697).
- Risk of bias appraisals using SYRCLE and RoB 2 tools were not applied strictly. Most studies were marked as "unclear" or "high risk," but authors ignored these outcomes in interpreting the data.
Answer: We appreciate the reviewer's comments. We want to clarify that the SYRCLE and RoB 2 tools were employed in strict accordance with their methodological guidelines. The high prevalence of “unclear” or “high risk” designations largely reflects the incomplete or inconsistent reporting evident in the original studies, particularly among preclinical investigations. In line with SYRCLE's recommendations, when sufficient information to assess a domain is lacking, the "unclear" designation is favored over making assumptions towards low or high risk. This conservative strategy was adopted to maintain transparency and reproducibility in our risk of bias evaluations.
We agree, however, that a more explicit discussion of the implications of these assessments was warranted in our findings. To address this, we have expanded our discussion to highlight that the frequent occurrence of unclear risk of bias, especially in critical domains such as randomization, blinding, and allocation concealment, poses a significant threat to the internal validity of many studies and compromises the reliability of the reported outcomes. This limitation is particularly pronounced for behavioral, histological, and mechanistic endpoints, which are inherently vulnerable to observer bias. Additionally, these methodological issues hinder the ability to compare effect sizes or determine the most effective formulations accurately.
For your review, please check the modifications marked in green in the revised manuscript, in the topic “Expert Opinion and Perspectives” (Manuscript’s page 30, Line 1224 to 1246).
- Studies covered are very heterogeneous in terms of curcumin forms, type of nanocarriers, routes of administration, and model animals. However, heterogeneity is not supported by the review without any subgroup analysis, normalization of results, or weightage based on relevance.
Answer: We appreciate the reviewer’s valuable observation. It is important to clarify that this study was conducted as a scoping review, aimed primarily at mapping the landscape and characteristics of existing research, rather than quantitatively synthesizing or ranking interventions by efficacy. In accordance with the Joanna Briggs Institute and PRISMA-ScR guidelines, scoping reviews are not designed to require subgroup analyses, normalization of outcomes, or comparative weighting across studies due to their exploratory focus.
Then, we conclude that such analyses could yield insightful outcomes and are indeed more suitable for systematic reviews or meta-analyses that address more specific research questions. To enhance our methodological rigor, we implemented a structured risk of bias assessment (utilizing SYRCLE and RoB 2), despite not being a requirement of current guidelines for scoping reviews. We hope the reviewer understands that our objective was to create a comprehensive evidence map to inform future research initiatives. As the field evolves and more homogeneous, higher-quality studies emerge, we anticipate pursuing focused systematic reviews, incorporating subgroup analyses or meta-analyses that target specific nanocarriers, curcumin derivatives, or delivery routes.
- The review is mostly descriptive with a focus on reporting formulation characteristics such as zeta potential and particle size and less on identifying if these correlate with efficacy, safety, or biological activities.
Answer: We appreciate the reviewer’s insightful and technically pertinent comments. We conclude that investigating the correlation between physicochemical attributes and biological outcomes is a critical element of formulation science. However, in the context of this scoping review, we were unable to systematically analyze such correlations due to several key limitations:
- Incomplete Reporting: Most of the studies included in our review lacked comprehensive analyses linking formulation’s characteristics to biological or therapeutic outcomes. While parameters like particle size and zeta potential were frequently reported, few studies assessed their influence on drug release kinetics, tissue penetration, or pharmacodynamic responses in a cohesive manner.
- High Methodological Heterogeneity: The included studies varied significantly in terms of animal models used (e.g., rats, mice, rabbits, chicks), administration routes (e.g., oral, intra-articular, topical, intravenous), dosing regimens, and outcome measures (such as biochemical markers, behavioral assays, and histological evaluations). This methodological diversity precludes meaningful cross-study comparisons and makes it difficult to draw causal inferences regarding the relationship between nanocarrier properties and biological activity.
- Lack of Standardization and Elevated Risk of Bias: As highlighted in previous sections, many studies exhibited methodological shortcomings. The absence of standardized efficacy endpoints, coupled with an unclear or elevated risk of bias in numerous domains, significantly undermines the reliability of any indirect comparisons.
To address these limitations, we have now incorporated a dedicated paragraph in the discussion that clarifies why a correlative analysis between nanocarrier characteristics and biological outcomes was infeasible in the context of this review.
For your review, please check the modifications marked in green in the revised manuscript, in the topic “Expert Opinion and Perspectives” (Manuscript’s page 31).
- The review doesn't state any gap-driven research or hypothesis questions. It doesn't provide suggestions for future directions of research or specific nanocarrier systems to be studied in the future.
Answer: We appreciate the reviewer’s comment. While scoping reviews primarily serve a descriptive function without the intention of formulating or testing specific hypotheses, they play a crucial role in identifying evidence gaps and outlining avenues for future research. In our study, we employed evidence mapping technique, utilizing a Sankey diagram to illustrate the distribution of various nanoformulation types, administration routes, experimental models, and study designs. This visualization not only highlighted areas of concentrated evidence but also revealed underexplored domains, such as semisolid formulations, topical administration routes, and advanced nanocarrier systems like exosomes and hybrids.
However, we recognize that the gaps identified, and future directions were not synthesized into a distinct section. To enhance clarity and address the reviewer’s feedback, we have revised the manuscript to include a comprehensive discussion of the research gaps highlighted by the Sankey mapping. We propose targeted recommendations for future investigations, which encompass the expansion of clinical studies, an emphasis on safety evaluations, exploration of alternative delivery mechanisms, and improvements in methodological standardization to facilitate translational advancements in CUR nanotherapeutics for arthritis.
These modifications have been marked in green in the revised manuscript (Manuscript’s page 32, Line 1311 to 1327).
- The English language requires further improvement.
Answer: We appreciate the reviewer’s observation. A thorough revision of the language has been conducted to enhance clarity, fluency, and overall readability of the manuscript. We reviewed grammatical structures, word choices, and technical phrasing to ensure consistency and precision in our scientific communication. These modifications have been marked in green in the revised manuscript.
Reviewer 3 Report
Comments and Suggestions for Authors
This scope review systematically evaluates the preclinical and clinical evidence of curcumin (CUR) nanocarriers in the treatment of arthritis, and the topic has important clinical significance. The structure of the article is basically complete, but there are logical contradictions, methodological flaws, insufficient data analysis, chart quality issues, and language errors that require significant revision.
- The SYRCLE tool evaluated preclinical studies (Figure 5), which showed that most areas were "uncertain risks", but the discussion section did not analyze their impact on the conclusions
- The Sankey diagram has overlapping text and unclear flow paths. Suggest simplifying the data
- The SYRCLE tool evaluates preclinical studies and shows that most areas have "uncertain risks," but the discussion section does not analyze its impact on conclusions (page 13).
- There are multiple grammar errors in the article, please carefully check them.
Round 2
Reviewer 3 Report
Comments and Suggestions for Authors
now the revised ms can be accepted.